# OFFLINE REINFORCEMENT LEARNING WITH COMBINATORIAL ACTION SPACES

## ABSTRACT

Reinforcement learning problems often involve large action spaces arising from the simultaneous execution of multiple sub-actions, resulting in combinatorial action spaces. Learning in combinatorial action spaces is difficult due to the exponential growth in action space size with the number of sub-actions and the dependencies among these sub-actions. In offline settings, this challenge is compounded by limited and suboptimal data. Current methods for offline learning in combinatorial spaces simplify the problem by assuming sub-action independence. We propose Branch Value Estimation (BVE), which effectively captures sub-action dependencies and scales to large combinatorial spaces by learning to evaluate only a small subset of actions at each timestep. Our experiments show that BVE outperforms state-of-the-art methods across a range of action space sizes.[1]

## 1 INTRODUCTION

Offline reinforcement learning (RL) automates sequential decision-making in domains where trial-and-error exploration is costly, risky, or impractical by learning from a fixed dataset (Lange et al., 2012). While effective in various domains (Fu et al., 2020; Levine et al., 2020), value-based offline RL methods often require exhaustive enumeration of the action space, and policy-based methods are typically designed for continuous action spaces (Lillicrap et al., 2016; Delarue et al., 2020). However, in many real-world settings, the concurrent execution of multiple actions creates large, discrete *combinatorial* action spaces, rendering traditional offline RL approaches ineffective. In healthcare, for example, practitioners must choose from thousands of procedural combinations at every decision point. Yet, to minimize risks and costs, they must only take the actions most informative for disease diagnosis and treatment, a notoriously difficult task (Yoon et al., 2019).

Learning in combinatorial action spaces is challenging due to the exponential increase in possible actions with action space dimensionality. In an $N$-dimensional action space with $m_d$ discrete sub-actions per dimension $d$, the total number of possible actions is given by $\prod_{d=1}^{N} m_d$. In traffic light control (Rasheed et al., 2020), for instance, where each light represents a dimension in the action space and its status (red, green, yellow) is a sub-action, controlling just four intersections with four lights each results in $3^{16}$ (>43M) possible actions. People naturally eliminate most unsuitable actions, such as turning all lights green simultaneously, using common sense. RL agents lack this intuition and must spend time and computational resources to discover the sub-optimality of nearly all action combinations (Zahavy et al., 2018). Although offline RL methods can learn to avoid ineffective actions through expert demonstrations (Levine et al., 2020), we find that state-of-the-art approaches struggle to resolve the complex dependencies among sub-actions, where the utility of one sub-action can critically depend on the presence or absence of another.

We introduce Branch Value Estimation (BVE) to learn in environments with discrete, combinatorial action spaces. Our key insight is that structuring combinatorial action spaces as trees can capture dependencies among sub-actions while reducing the number of actions evaluated at each timestep. Specifically, in our action space tree (Figure 1), each node represents a distinct sub-action combination, and each edge assigns a unique value to a specific sub-action. The tree is structured so that a node inherits the values of sub-actions from its ancestors, with siblings having distinct values for

---

[1]Our implementation is available at `https://anonymous.4open.science/r/branch_value_estimation-B911`

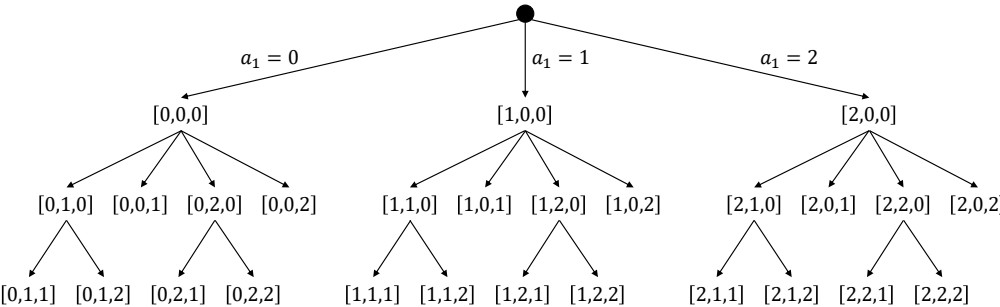

Figure 1: Consider an action space tree for a three-dimensional action $\mathbf{a} = [a_1, a_2, a_3]$ with each $a_i \in \{0, 1, 2\}$. Each node represents a unique sub-action combination, and edges assign values to the current sub-action combination. Nodes inherit values from ancestors, with siblings differing only in the current sub-action. For instance, at the first level, sibling nodes $[0, 0, 0]$, $[1, 0, 0]$, and $[2, 0, 0]$ differ in $a_1$. In the subtree rooted at $[1, 0, 0]$, all descendant nodes have $a_1 = 1$, with variations occurring in the subsequent dimensions $a_2$ and $a_3$.

the sub-action currently under consideration. At each tree level, BVE identifies the optimal sub-action value by estimating the highest achievable Q-value conditioned on each value in $m_d$ being assigned to the sub-action. This traversal process continues until a complete action is constructed, which is then used for learning via a behavior-regularized TD loss function. After training, we use beam search (Reddy, 1977) to traverse the action space tree and extract the optimal action at each timestep. BVE outperforms state-of-the-art baselines in environments with action spaces ranging from 16 to over 4 million actions, as illustrated for the largest space in Figure 2.

Our contributions are as follows:

1. We define a behavior-regularized TD loss function that inherently captures dependencies among sub-actions in discrete combinatorial action spaces.

2. We introduce BVE, an offline RL method for learning in discrete, combinatorial action spaces. BVE handles sub-action dependencies and scales to large action spaces by representing the action space as a tree. At each timestep, BVE selects the optimal action by traversing the tree and predicting the maximum Q-value achievable along each branch.

3. Our experiments demonstrate that BVE consistently outperforms state-of-the-art baselines in discrete, combinatorial action spaces, regardless of action space size or sub-action dependencies.

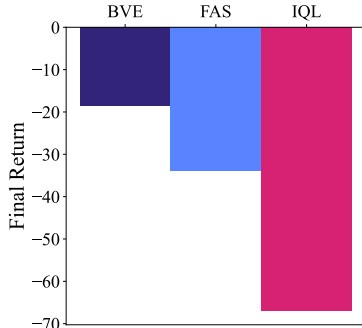

Figure 2: BVE outperforms state-of-the-art methods in complex, combinatorial action spaces.

## 2 PRELIMINARIES

Reinforcement learning problems can be formalized as a Markov Decision Process (MDP), $\mathcal{M} = \langle \mathcal{S}, \mathcal{A}, p, r, \gamma, \mu \rangle$ where $\mathcal{S}$ is a set of states, $\mathcal{A}$ is a set of actions, $p : \mathcal{S} \times \mathcal{A} \times \mathcal{S} \to [0, 1]$ is a function that gives the probability of transitioning to state $s'$ when action $a$ is taken in state $s$, $r : \mathcal{S} \times \mathcal{A} \to \mathbb{R}$ is a reward function, $\gamma \in [0, 1]$ is a discount factor, and $\mu : \mathcal{S} \to [0, 1]$ is the distribution of initial states. A policy $\pi : \mathcal{S} \to \mathbb{P}(\mathcal{A})$ is a distribution over actions conditioned on a state $\pi(a \mid s) = \mathbb{P}[a_t = a \mid s_t = s]$. In our work, we assume states $\mathcal{S}$ can be either discrete or continuous and that the MDP has a finite horizon $H$.

While the standard MDP formulation abstracts away the structure of actions in $\mathcal{A}$, we explicitly assume that the action space is combinatorial; that is, $\mathcal{A}$ is defined as a Cartesian product of sub-action spaces. More formally, $\mathcal{A} = \mathcal{A}_1 \times \mathcal{A}_2 \times \cdots \times \mathcal{A}_N$, where each $\mathcal{A}_d$ is a discrete set. Consequently, $\mathbf{a}_t$ is an $N$-dimensional vector wherein each component is referred to as a sub-action.

The agent's goal is to learn a policy $\pi^*$ that maximizes cumulative discounted returns:

$$\pi^* = \arg\max_\pi \mathbb{E}_\pi \left[ \sum_{t=0}^H \gamma^t r(s_t, a_t) \mid s_0 \sim \mu(\cdot), a_t \sim \pi(\cdot \mid s_t), s_{t+1} \sim p(\cdot \mid s_t, a_t) \right] .$$

In online RL, an agent learns by trial and error interaction with its environment. In offline RL, by contrast, the agent learns from a static dataset of transitions $\mathcal{B} = \{(s_t, a_t, r_t, s_{t+1})^i\}_{i=0}^N$ generated by, possibly, a mixture of policies collectively referred to as the behavior policy $\pi_\beta$.

Like many recent offline RL methods, our work uses approximate dynamic programming to minimize temporal difference error (TD error) starting from the following loss function:

$$L(\theta) = \mathbb{E}_{(s,a,r,s')\sim\mathcal{B}} \left[ \left( r + \gamma \max_{a'} Q(s', a'; \theta^-) - Q(s, a; \theta) \right)^2 \right] , \qquad (1)$$

where $Q(s, a; \theta)$ is a parameterized Q-function that estimates the expected return when taking action $a$ in state $s$ and following the policy $\pi$ thereafter, and $Q(s, a; \theta^-)$ is a target network with parameters $\theta^-$, which is used to stabilize learning.

For out-of-distribution actions $a'$, Q-values can be inaccurate, often causing overestimation errors due to the maximization in equation 1. To mitigate this effect, offline RL methods either assign lower values to these out-of-distribution actions via regularization or directly constrain the learned policy. For example, TD3+BC (Fujimoto & Gu, 2021) adds a behavior cloning term to the standard TD3 loss:

$$\pi = \arg\max_\pi \mathbb{E}_{(s,a)\sim\mathcal{B}} \left[ \lambda Q(s, \pi(s)) - (\pi(s) - a)^2 \right] , \qquad (2)$$

where $\lambda$ is a scaling factor that controls the strength of the regularization.

More recently, implicit Q-learning (IQL) (Kostrikov et al., 2021) used a SARSA-style TD backup and expectile loss to perform multi-step dynamic programming without evaluating out-of-sample actions:

$$L(\theta) = \mathbb{E}_{(s,a,r,s')\sim\mathcal{B}} \left[ \left( r + \gamma \max_{a'\in\Omega(s)} Q(s', a'; \theta^-) - Q(s, a; \theta) \right)^2 \right] , \qquad (3)$$

where $\Omega(s) = \{a \in A \mid \pi_\beta(a \mid s) > 0\}$ are actions in the support of the data.

As we will describe in section 3, we combine ideas from TD3+BC (equation 2) and IQL (equation 3) to create a regularized, SARSA-style TD loss function.

## 3 BRANCH VALUE ESTIMATION

Learning near-optimal policies in discrete, combinatorial action spaces often requires accounting for dependencies among sub-actions. We thus create a TD loss function that is defined across all action dimensions:

$$L_{TD}(\theta) = \mathbb{E}_{(s,\mathbf{a},r,s',\mathbf{a}')\sim\mathcal{B}} \left[ \left( r + \gamma \left( \lambda Q(s', \hat{\mathbf{a}}'; \theta^-) - \|\hat{\mathbf{a}}' - \mathbf{a}'\| \right) - Q(s, a; \theta) \right)^2 \right] , \qquad (4)$$

where $\hat{\mathbf{a}}'$ is $\arg\max_{a'} Q(s', a'; \theta^-)$ in equation 1.

This loss inherently captures dependencies among sub-actions by evaluating actions as integrated wholes rather than as aggregates of their individual components, such as in a linear decomposition (Tang et al., 2022). As the action space grows exponentially with the number of sub-actions, traditional value-based RL methods struggle to accurately identify $\hat{\mathbf{a}}'$ due to errors in Q-function estimation. These errors frequently result in convergence to suboptimal policies, especially in environments with large action spaces (Thrun & Schwartz, 1993; Zahavy et al., 2018). Our experiments, detailed in section 4.3, corroborate these findings.

To overcome this phenomenon, we create an action space tree wherein each node represents a unique combination of sub-actions, and each edge assigns a specific value to a sub-action in $\mathbf{a}_t$. A node inherits previously assigned sub-action values from its ancestors, while its siblings have distinct values for the sub-action currently under consideration (Figure 1). We impose no restrictions on

sub-action cardinalities. However, for clarity, subsequent examples will focus on multi-binary action spaces, where sub-actions are either included ($a_i = 1$) or excluded ($a_i = 0$).

To determine the optimal action $\hat{\mathbf{a}}'$, we traverse the action space tree with a neural network $f : \mathbb{R}^{|\mathcal{S}| \times |\mathcal{A}|} \rightarrow \mathbb{R}^{1 \times |\mathcal{A}_d|}$, parameterized by $\theta$. This network predicts a node's scalar Q-value $q$ and a vector of *branch values* $\mathbf{v}$, where $(q, \mathbf{v}) = f(s, \mathbf{a}; \theta)$. Each $v_i \in \mathbf{v}$ represents the maximum Q-value reachable through the sub-tree rooted at its corresponding child node.

Let $\mathbf{u} = [q, v_1, v_2, \ldots, v_m]$ denote a vector comprising the predicted scalar Q-value $q$ and the branch values $\mathbf{v}$ for the given $(s, \mathbf{a})$. Each component $u_i$ represents the value of selecting its corresponding node. Tree traversal proceeds to nodes with probability proportional to their values:

$$\pi(u_i \mid s) = \frac{\exp\left(u_i / \tau\right)}{\sum_{j=0}^{m} \exp\left(u_j / \tau\right)} \, ,$$

where $\tau$ is the temperature parameter.

Traversal terminates under two conditions. First, if a leaf node is reached, meaning every sub-action has been explicitly assigned a value. Second, if a node's Q-value exceeds all of its children's branch values. This second condition ensures that the agent can access every action, not just those with a specific number of sub-actions. For instance, in the action space illustrated in Figure 1, the agent must be able to select any of the 27 actions in each state. If the agent is constrained to traverse to a leaf rather than selecting an action where $q > v_i \forall v_i \in \mathbf{v}$, some actions, such as [1,0,0], would be unavailable. BVE's tree traversal procedure is illustrated in Figure 3.

The parameters $\theta$ of our network $f(s, \mathbf{a}; \theta)$ are updated to minimize regularized TD error (equation 4) and branch value error $L = \alpha L_{TD} + L_{BVE}$, where $\alpha$ adjusts the contribution of the TD loss to the total loss. Branch value error is computed starting from a node $\mathbf{a}$ sampled from $\mathcal{B}$, with a target defined by equation 4. The target is propagated to $\mathbf{a}$'s parent node, where it is used to compute loss and is then updated to the maximum of the propagated target and the branch values of the parent's other children. As shown in Algorithm 1 and Figure 4, this process repeats until the loss for all nodes is computed.

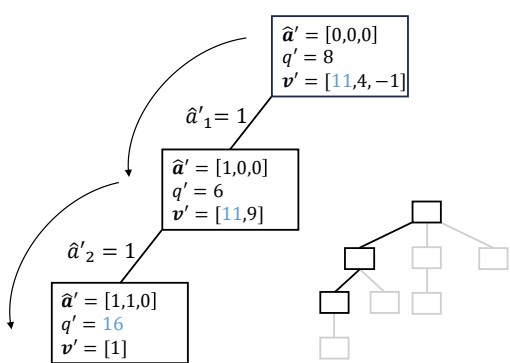

Figure 3: BVE traversal when $\mathbf{a} \in \{0, 1\}^3$ (with the full action space tree at bottom-right). Starting from the root node $\hat{\mathbf{a}}' = [0, 0, 0]$, we select $\hat{a}_1' = 1$ as its branch value (11) exceeds the root's Q-value (8) and the other children's branch values (4 and $-1$). Traversal continues, including $\hat{a}_2' = 1$, to $\hat{\mathbf{a}}' = [1, 1, 0]$, which is chosen because its Q-value (16) is greater than its child's branch value (1).

While the behavior cloning regularizer in equation 4 minimizes overestimation error, further mitigation is possible by sparsifying the action space tree to include only actions in $\mathcal{B}$ (Fujimoto et al., 2019). Leveraging the behavior policy's expertise in this manner is particularly advantageous in real-world settings where some sub-actions never co-occur, leaving a much smaller subset of viable action combinations. For example, in healthcare, certain medications are never simultaneously prescribed due to their conflicting effects.

We use two methods to reduce errors in action selection caused by inaccurate branch value estimations near the tree root. First, we introduce a depth penalty parameter, $\delta$, to weigh the contribution of nodes during the BVE loss calculation. Because we traverse from node to root, $\delta \geq 1$ assigns greater weight to branch value errors closer to the root, prioritizing corrections at higher levels of the tree, where decisions have a broader impact on the selected action (see line 12 of Algorithm 1). Second, when extracting a policy after learning, we use beam search (Reddy, 1977), a technique from natural language processing, to enable a broader exploration of action combinations. Specifically, we use the same tree traversal process illustrated in Figure 3, except we retain the top $W$ actions — based on their values in $\mathbf{u}$ — at each level for further exploration. The best action from all explored beams is selected at the end of the search.

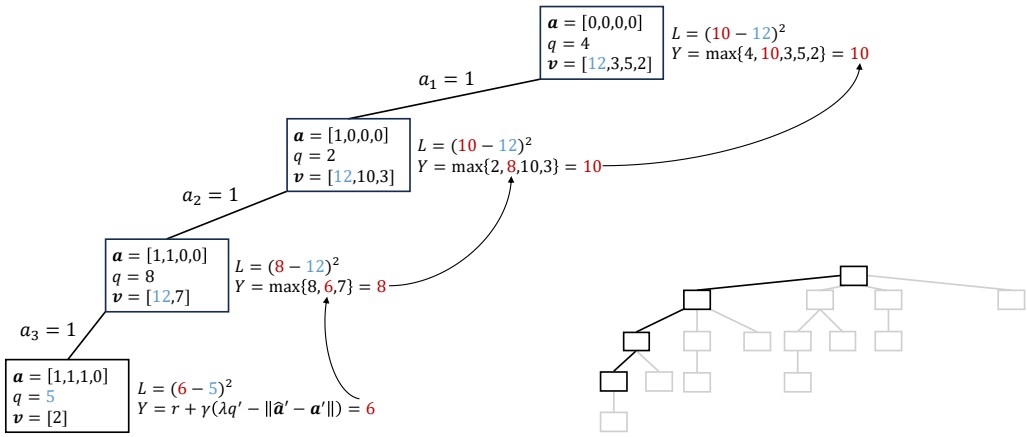

Figure 4: In this example, $\mathbf{a} \in \{0,1\}^4$ (full action space tree at bottom-right). We calculate branch value error starting from the sampled node $\mathbf{a} = [1, 1, 1, 0]$, using a target defined by equation 4. This target is propagated to the parent node $\mathbf{a} = [1, 1, 0, 0]$. At this parent node, the target is determined by taking the maximum between the propagated target and the branch values of the node's other children. This new target is then propagated up the tree. The process repeats until the loss for all nodes is calculated.

In summary, BVE learns in discrete combinatorial action spaces by estimating Q-values using equation 4. Unlike traditional RL methods, which often misidentify the optimal next action $\hat{\mathbf{a}}'$ in equation 4, BVE reduces the effective action space by organizing it as a tree. The optimal action, $\hat{\mathbf{a}}'$, is found through a traversal process, guided by a neural network that predicts each node's scalar Q-value and a vector of branch values. Each branch value represents the maximum Q-value attainable from the sub-tree rooted at the corresponding child node. The network is updated by minimizing a weighted sum of TD loss (equation 4), which is a behavior-regulaized variant of the standard RL loss, and the BVE loss (Algorithm 1), which reduces branch value prediction errors.

## 4 EXPERIMENTAL EVALUATION

We evaluate the effectiveness of BVE in an $N$-dimensional grid world in which each sub-action corresponds to movement in a specified direction. For example, in a 2D grid, the agent can move in directions defined by combinations of up ($U$), down ($D$), right ($R$), and left ($L$) (e.g., $[U]$, $[UR]$, $[UDL]$, $[UDRL]$, etc.). Opposing sub-actions (e.g., $[UD]$) cancel each other out when selected simultaneously, whereas complementary sub-actions (e.g., $[UR]$) enable the agent to reach the goal more efficiently than executing the same actions sequentially (e.g., $a_{t_1} = [U]$, $a_{t_2} = [R]$). Notably, the complexity of this environment grows exponentially with $N$, as both the action space ($2^{2N}$) and state space ($K^N$, where $K$ is the grid size) scale with the grid dimension.

At each timestep, the agent receives a negative reward $-\rho(s, g)$ proportional to its distance from the goal, except in the goal state or a pit. The goal and pit states are terminal, with a pit being associated with failure. Upon reaching the goal, the agent receives $r = 10$. Because the agent incurs a negative

---

**Algorithm 1** Compute BVE Loss

**Require:**
    $f(\theta)$: neural network with parameters $\theta$
    $f(\theta^-)$: target network with parameters $\theta^-$
    $\{s, \mathbf{a}, r, s', \mathbf{a}'\}$: transition from $\mathcal{B}$
    $\hat{\mathbf{a}}'$: action selected via tree traversal given $s$
1: $(q, \mathbf{v}) \leftarrow f(s, \mathbf{a}; \theta)$
2: $(q', \mathbf{v}') \leftarrow f(s', \hat{\mathbf{a}}'; \theta^-)$
3: $Y \leftarrow r + \gamma(\lambda q' - \|\hat{\mathbf{a}}' - \mathbf{a}'\|)$
4: total loss $\leftarrow (q - Y)^2$
5: node $\leftarrow \mathbf{a}$
6: $d \leftarrow 1$
7: **while** node is not *null* **do**
8:     parent $\leftarrow$ GETPARENT(node)
9:     $q, \mathbf{v} \leftarrow f(s, \text{parent}; \theta)$
10:     children $\leftarrow$ GETCHILDREN(parent)
11:     $i \leftarrow$ index of node in children
12:     loss $\leftarrow ((\mathbf{v}[i] - Y) * \delta d)^2$
13:     total loss $\leftarrow$ total loss + loss
14:     $\mathbf{v}[i] \leftarrow Y$
15:     $Y \leftarrow \max(q, \mathbf{v})$
16:     node $\leftarrow$ parent
17:     $d \leftarrow d + 1$
18: **end while**
19: **return** total loss$/d$

---

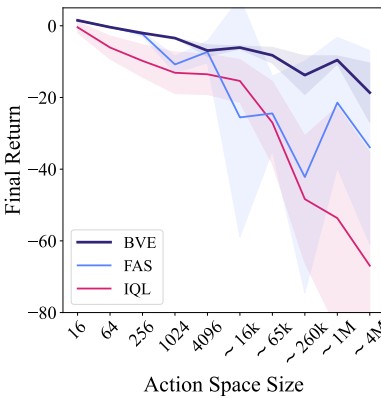

| $|\mathcal{A}|$ | BVE | FAS | IQL |
|---|---|---|---|
| 16 | 1.5 $\pm 0.0$ | 1.5 $\pm 0.0$ | $-0.4 \pm 1.5$ |
| 64 | -0.4 $\pm 0.0$ | -0.4 $\pm 0.0$ | $-6.1 \pm 3.2$ |
| 256 | -2.0 $\pm 0.1$ | $-2.3 \pm 0.6$ | $-9.8 \pm 4.5$ |
| 1024 | -3.4 $\pm 0.1$ | $-10.8 \pm 2.3$ | $-13.1 \pm 5.8$ |
| 4096 | -6.9 $\pm 1.6$ | $-7.3 \pm 3.1$ | $-13.5 \pm 5.7$ |
| $\sim$16k | -6.1 $\pm 0.4$ | $-25.6 \pm 33.3$ | $-15.4 \pm 6.0$ |
| $\sim$65k | -8.2 $\pm 2.2$ | $-24.4 \pm 10.4$ | $-27.0 \pm 11.5$ |
| $\sim$260k | -13.8 $\pm 5.4$ | $-42.2 \pm 32.3$ | $-48.4 \pm 17.7$ |
| $\sim$1M | -9.6 $\pm 1.2$ | $-21.4 \pm 18.2$ | $-53.7 \pm 31.0$ |
| $\sim$4M | -18.6 $\pm 8.3$ | $-33.9 \pm 27.0$ | $-66.9 \pm 31.6$ |

Figure 5: Average returns and standard deviations calculated from the final 15 evaluations and 5 seeds. The best results are highlighted in blue . In lower-dimensions, FAS matches BVE's performance. However, in higher dimensional environments, the discrepancy between the linearly decomposed and true reward functions becomes more significant, leading to instability in FAS's learning.

reward at each timestep, it may be incentivized to enter a pit if reaching the goal requires covering a long distance. To deter this behavior, a penalty ten times the distance from the agent's starting location to the goal ($r = -10 * \rho(s_0, g)$) is imposed for falling into a pit.

In this deterministic grid-world domain, we use an augmented form of A$^*$ to generate our dataset $\mathcal{B}$. Because the optimal policy requires few actions to reach the goal, the A$^*$ agent selects the optimal action with a probability of 0.1, choosing randomly otherwise to ensure state-action diversity in $\mathcal{B}$.

**Baseline Comparison** We compare BVE's performance to state-of-the-art baselines, Factored Action Spaces (FAS) (Tang et al., 2022), which learns linearly decomposable Q-functions for combinatorial action spaces, and Implicit Q-Learning (IQL) (Kostrikov et al., 2021), a general-purpose offline RL method included to demonstrate the necessity of approaches purpose-built for combinatorial action spaces. We train each algorithm for 20,000 gradient steps, assessing the learned policy every 100 timesteps.

**Experimental Setup** We evaluate these methods in 20 environments, categorized into two types: those with and without a cluster of pits along the optimal path. We create ten instances for each type, varying in dimension from 2D, with 16 available actions in each state ($|\mathcal{A}| = 16$) (i.e., $\{\emptyset, [U], [UD], [UDL], [ULR], [UDLR], [D], [DR], \dots\}$), to 11D, with over four million available actions in each state ($|\mathcal{A}| = 4{,}194{,}304$). We use a grid of size 5 in each dimension. Consequently, the smallest environment, in 2D, has 25 states, while the largest, in 11D, exceeds 48 million states. In all environments, the agent begins in the bottom left corner and the goal state is in the top right corner. We present results averaged over five seeds, with the shaded areas in our figures indicating one standard deviation.

## 4.1 N-Dimensional Grid World Without Pits

In the pit-free environments, the agent's task is relatively simple because the optimal action is the same in all states. Moreover, the transition probability from $s$ to $s'$ can be decomposed into independent probabilities for each sub-action, and the policy into a product of independent sub-action policies.

Notably, sub-actions aren't fully independent, as the reward model cannot be decomposed into separate rewards. Still, FAS learns a high-performing policy despite the bias from its linear decomposition, as sub-action interactions are relatively mild. In higher-dimensional environments, however, the difference between the linearly decomposed and true reward functions becomes more pronounced, causing instability in FAS's learning. BVE, by contrast, does not exhibit this behavior, as our loss (equation 4) evaluates actions as unified entities rather than aggregates of individual components.

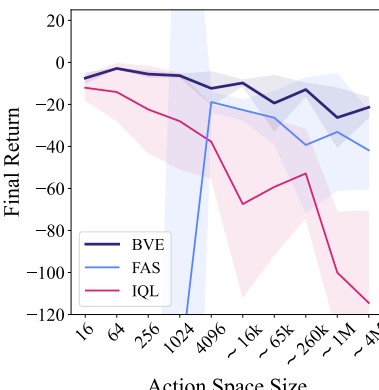

| $|\mathcal{A}|$ | BVE | FAS | IQL |
|---|---|---|---|
| 16 | -7.5 $\pm$ 2.4 | $-531.5 \pm 31.4$ | $-12.1 \pm 5.8$ |
| 64 | -2.9 $\pm$ 0.2 | $-579.8 \pm 7.2$ | $-14.1 \pm 13.6$ |
| 256 | -5.6 $\pm$ 1.5 | $-480.3 \pm 152.9$ | $-22.4 \pm 20.6$ |
| 1024 | -6.3 $\pm$ 0.7 | $-147.4 \pm 292.6$ | $-28.0 \pm 22.8$ |
| 4096 | -12.4 $\pm$ 7.9 | $-18.9 \pm 4.6$ | $-37.8 \pm 17.4$ |
| $\sim$16k | -9.8 $\pm$ 1.5 | $-22.6 \pm 4.8$ | $-67.4 \pm 44.2$ |
| $\sim$65k | -19.4 $\pm$ 13.2 | $-26.3 \pm 12.5$ | $-59.3 \pm 32.2$ |
| $\sim$260k | -12.9 $\pm$ 3.0 | $-39.3 \pm 32.0$ | $-52.9 \pm 21.0$ |
| $\sim$1M | -26.3 $\pm$ 14.1 | $-33.1 \pm 27.7$ | $-100.0 \pm 28.7$ |
| $\sim$4M | -21.4 $\pm$ 4.7 | $-41.8 \pm 18.4$ | $-114.5 \pm 43.7$ |

Figure 6: Average returns and standard deviations calculated from the final 15 evaluations and 5 seeds. The best results are highlighted in blue . BVE outperforms both FAS and IQL across all environments. FAS struggles in lower dimensions due to the stronger dependencies among sub-actions in these settings, performing poorly until $|\mathcal{A}|$ =4,096.

Because BVE explicitly accounts for interactions between sub-actions, it performs as well as or better than FAS and IQL, as demonstrated in Figure 5. Full learning curves for these experiments are available in Appendix A.

## 4.2 N-DIMENSIONAL GRID WORLD WITH PITS

We create pit clusters by placing a pit on the optimal path and randomly adding four additional adjacent pits, thus ensuring the optimal policy requires a diverse set of actions with varying numbers of sub-actions.

In worlds with pits, action effectiveness critically depends on sub-action coordination, especially in lower-dimensional environments. For example, in two dimensions, navigating around a pit requires careful selection of all sub-actions. Because the number of states grows exponentially with dimensionality, higher-dimensional environments offer more paths for an agent to navigate around a pit. Consequently, lower-dimensional environments are higher-stakes; the wrong combination of just two actions can doom the agent. This complexity explains why FAS underperforms in lower dimensions, while BVE performs well in all worlds as shown in Figure 6. Full learning curves for these experiments are provided in Appendix A.

## 4.3 ABLATIONS AND HYPERPARAMETERS

As described in section 3, we apply a depth penalty $\delta$ to minimize action selection errors due to inaccurate branch value estimations near the tree root. This section evaluates the impact of removing this penalty. Additionally, because BVE learns through a weighted combination of TD loss (equation 4) and BVE loss (Algorithm 1), $L = \alpha L_{TD} + L_{BVE}$, we examine its sensitivity to $\alpha$. Finally, to assess the necessity of our tree structure, even with the inductive bias from selecting actions in $\mathcal{B}$ (section 3), we compare BVE's performance with that of a Deep Q-Network (DQN) (Mnih et al., 2015). The DQN is constrained to select actions from the dataset using its standard action-selection mechanism and is trained with BVE's TD loss function (equation 4). These experiments are conducted in environments with pits.

We observe that BVE shows minimal sensitivity to the depth penalty, set to $\delta = 1$ across all environments. However, as Figure 7a and Appendix B.1, illustrate, incorporating this penalty is crucial for both learning speed and asymptotic policy quality, especially as dimensionality increases.

BVE's performance remains stable over a large range of $\alpha$ values, particularly in lower-dimensional environments. In higher-dimensional settings, larger $\alpha$ values generally yield better results. Interestingly, in simpler, lower-dimensional environments, $\alpha = 0$ can still be effective. We hypothesize this is due to the inclusion of TD error in the BVE error calculation, as detailed in Algorithm 1

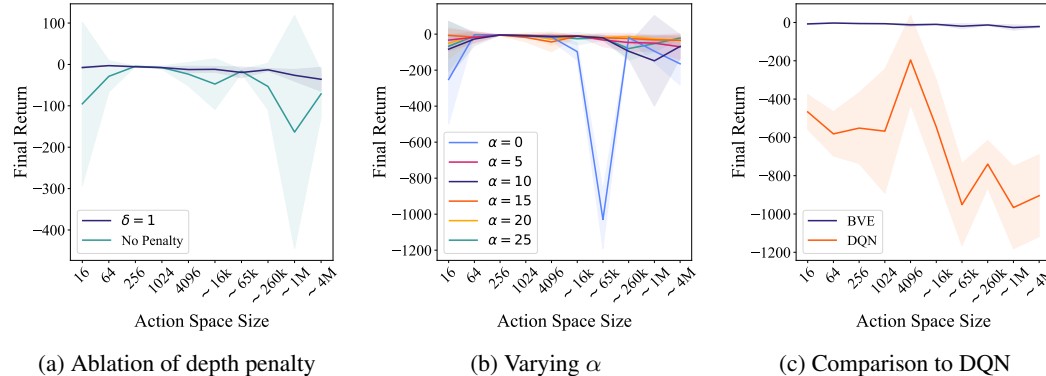

(a) Ablation of depth penalty      (b) Varying $\alpha$      (c) Comparison to DQN

Figure 7: Ablation study over BVE's components. While removing the depth penalty $\delta$ does not affect results in some environments, in others, it hurts performance considerably (Figure 7a). Performance remains stable across various $\alpha$ values, but removing TD loss from the total loss calculation ($\alpha = 0$) may result in sub-optimal policies (Figure 7b). Despite the inductive bias from constraining the DQN to select actions in $\mathcal{B}$, it performs poorly (Figure 7c).

and illustrated in Figure 4. However, omitting this term from the loss calculation can lead to catastrophic consequences, as observed in the 8D environment (Figure 7b). Full learning curves for these experiments are available in Appendix B.2.

Figure 7c and Appendix B.3 illustrate the tree structure's importance to BVE's effectiveness. Though trained with the same behavior-regularized TD loss function as BVE and restricted to selecting actions in $\mathcal{B}$, the DQN performs poorly. This indicates that the DQN struggles to manage the dependencies between sub-actions, particularly when there are many actions from which to choose. For instance, in the 11D world, the DQN must predict the 8,927 unique actions in $\mathcal{B}$ simultaneously. BVE mitigates this complexity by structuring the action space as a tree, thereby requiring predictions for only a small subset of Q-values at each timestep.

## 5 RELATED WORK

### 5.1 TREE-BASED RL

Monte Carlo Tree Search (MCTS) (Coulom, 2006), used most notably in AlphaZero (Silver et al., 2018), recursively selects actions using the Polynomial Upper Confidence Trees (PUCT) algorithm (Auger et al., 2013). PUCT selects action $a_t$ as $a_t = \mathrm{argmax}_a(Q(s_t, a) + U(s_t, a))$, where $U(s_t, a)$ provides an upper confidence bound on Q-values. Traditionally, this method is used for an *ordered* decision process, where the value of an action at time $t$ depends on subsequent actions at $t_1, t_2, \ldots, t_H$, as in chess. Therefore, MCTS is ill-suited for environments with unordered or *categorical* actions, like in our experiments, where sub-actions must be selected simultaneously.

TreeQN (Farquhar et al., 2017) integrates model-free RL with online planning by constructing an abstract MDP model that combines learned transition dynamics and reward predictions. It builds a tree of state representations and rewards for all action sequences up to a specified depth. Value estimates are recursively refined through a tree backup process to improve their accuracy.

Because traditional decision trees are non-differentiable if-then rules, they are incompatible with gradient descent, limiting their use in online RL. Silva et al. (2020) address this by introducing differentiable decision trees (DDTs), which replace rigid decision boundaries with smooth, differentiable functions, enabling gradient-based optimization in RL. After training, DDTs can be converted back into discrete trees, preserving interpretability.

Ernst et al. (2005) propose an offline RL approach that uses tree-based supervised learning algorithms within a fitted Q-iteration framework to approximate the Q-function. This method iteratively refines the Q-function using classical techniques like CART, Kd-trees, and tree bagging, leverag-

ing observed system transitions. By applying tree-based regression, the approach generalizes the learned policy to unobserved state-action pairs.

## 5.2 COMBINATORIAL ACTION SPACES

Due to the prevalence of combinatorial action spaces in real-world problems, various methods have been developed for learning in these environments. Many of these are tailored to specific domains, including text-based games and natural language action spaces (Zahavy et al., 2018; He et al., 2015; 2016), vehicle routing (Delarue et al., 2020; Nazari et al., 2018), the traveling salesperson problem (Bello et al., 2016), and resource allocation (Chen et al., 2024). These methods, however, often depend on problem-specific assumptions, whereas BVE is designed for broader applicability.

Other approaches are more general-purpose. For example, Tavakoli et al. (2018) introduce a novel architecture that distributes action controller representations across individual network branches, with a shared decision module encoding a latent input representation to coordinate these branches. Farquhar et al. (2020) propose using a curriculum of progressively expanding action spaces to accelerate learning in online environments where random exploration may be inefficient. This approach is effective when a restricted action space enables random exploration to generate significantly more informative experiences than in the full action space, and when regularities in the action space facilitate transferring learning to the full task. Amortized Q-learning (AQL) (Van de Wiele et al., 2020) avoids exact maximization over the action set at each step. Instead, it learns to search for the optimal action, thereby amortizing the cost of action selection over training. The search is treated as a distinct learning task, replacing exact maximization with maximization over a set of actions sampled from a learned proposal distribution.

While these methods are designed for online learning, Tang et al. (2022) propose an offline approach, which we refer to as FAS in our experiments, that linearly decomposes the Q-function, conditioning each component on a single sub-action and the full state space. This reduces the action space's dimensional complexity but the sufficient conditions for unbiased Q-value estimations — in effect, independence among sub-actions — often do not hold in real-world environments. BVE, by contrast, simplifies the problem by structuring the action space, enabling its application to problems where sub-actions may be interdependent.

## 6 CONCLUSION

In many real-world sequential decision making problems, discrete combinatorial action spaces emerge from the simultaneous selection of multiple sub-actions. Traditional RL approaches struggle in these spaces due to both the exponential increase in the action space size with the number of sub-actions and the complex dependencies among the sub-actions. These challenges are exacerbated in offline settings, where available data is often limited and sub-optimal. We present Branch Value Estimation (BVE), an offline RL method for learning in discrete, combinatorial action spaces. By structuring combinatorial action spaces as trees, BVE captures sub-action dependencies while reducing the number of actions evaluated per timestep, thus allowing it to scale to large action spaces. Our empirical experiments demonstrate that BVE outperforms state-of-the-art baselines across environments with varying action space sizes and sub-action dependencies. Future work should explore using BVE within an actor-critic framework to extend its applicability to continuous and mixed (discrete and continuous) combinatorial action spaces.

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

## A    N-DIMENSIONAL GRID WORLD LEARNING CURVES

We compare BVE's performance to state-of-the-art baselines, Factored Action Spaces (FAS) (Tang et al., 2022), which learns linearly decomposable Q-functions for offline combinatorial action spaces, and Implicit Q-Learning (IQL) (Kostrikov et al., 2021), a general-purpose offline RL method.

We evaluate these methods in 20 $N$-dimensional grid worlds in which each sub-action corresponds to movement in a specified direction. The complexity of these environments increases exponentially with $N$, with both the action space ($2^{2N}$) and state space ($K^N$, where $K$ is the grid size) scaling with the grid dimension.

The environments are categorized into two types: those with and without a cluster of pits along the optimal path. We create ten instances for each type, varying in dimension from 2D ($|\mathcal{A}| = 16$), to 11D ($|\mathcal{A}| = 4,194,304$). We use a grid of size 5 in each dimension, resulting in 25 states in the smallest (2D) environment and over 48 million states in the largest (11D) environment.

This section presents the learning curves for these methods, averaged over five seeds, with shaded areas representing one standard deviation across these seeds.

### A.1    WITHOUT PITS

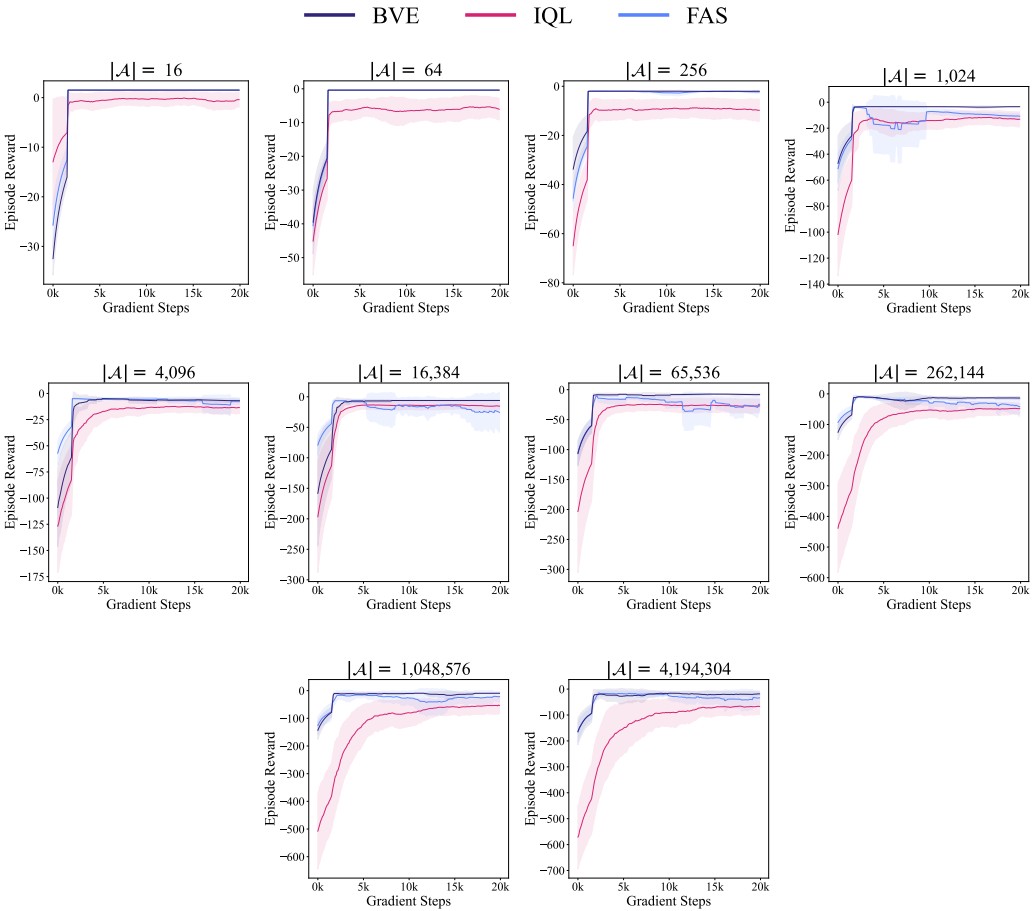

Figure 8: Learning curves for all agents in pit-less environments show that both BVE and FAS quickly establish effective policies; however, FAS exhibits instability when environmental dimensionality exceeds 4D ($|\mathcal{A}| = 256$).

## A.2  WITH PITS

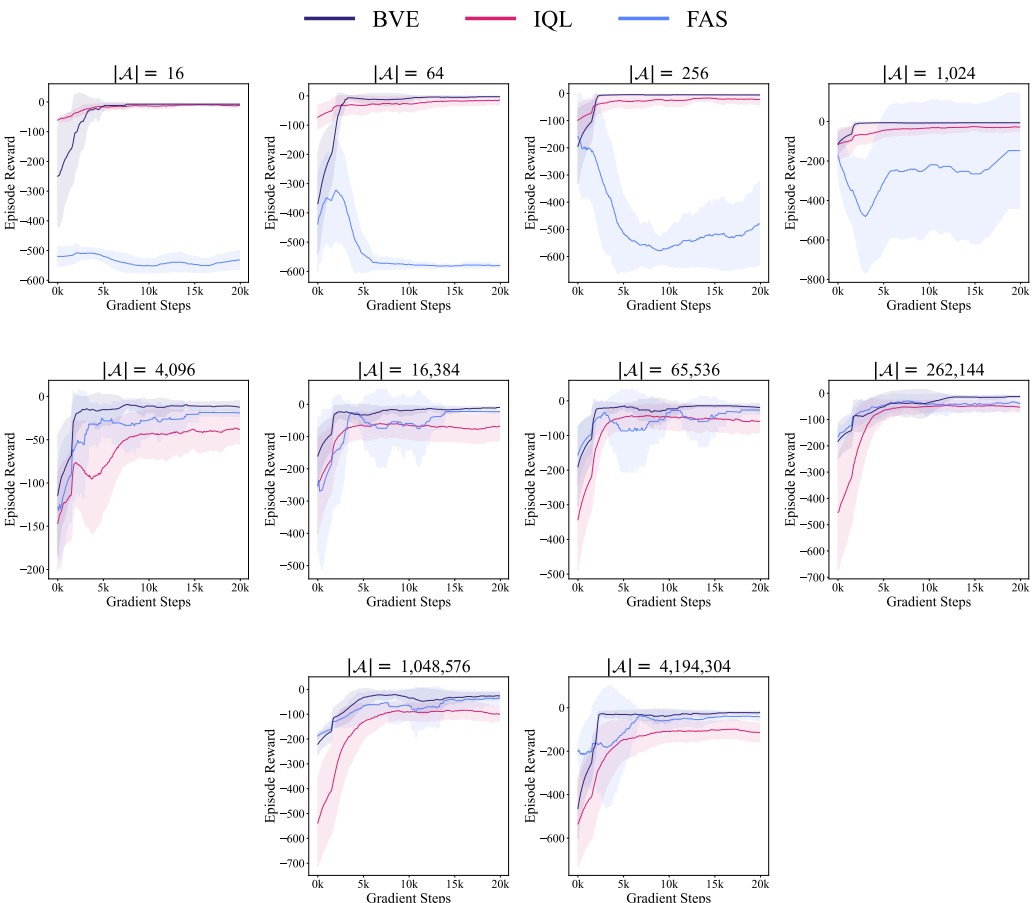

Figure 9: Learning curves in pit environments show BVE outperforming all baselines, especially FAS in lower dimensions where sub-actions are strongly dependent.

## B    ABALATION AND HYPERPARAMTER LEARNING CURVES

This section presents learning curves from three ablation/hyperparameter studies: 1) the impact of removing the depth penalty $\delta$, 2) BVE's sensitivity to $\alpha$, the weight of TD error in our total loss, and 3) the necessity of BVE's tree structure. In the third study, we compare BVE's performance to that of a DQN constrained to selecting actions from the dataset $\mathcal{B}$ and trained using BVE's TD loss function (equation 4). All experiments were conducted in environments with pits.

### B.1    DEPTH PENALTY ABALATION

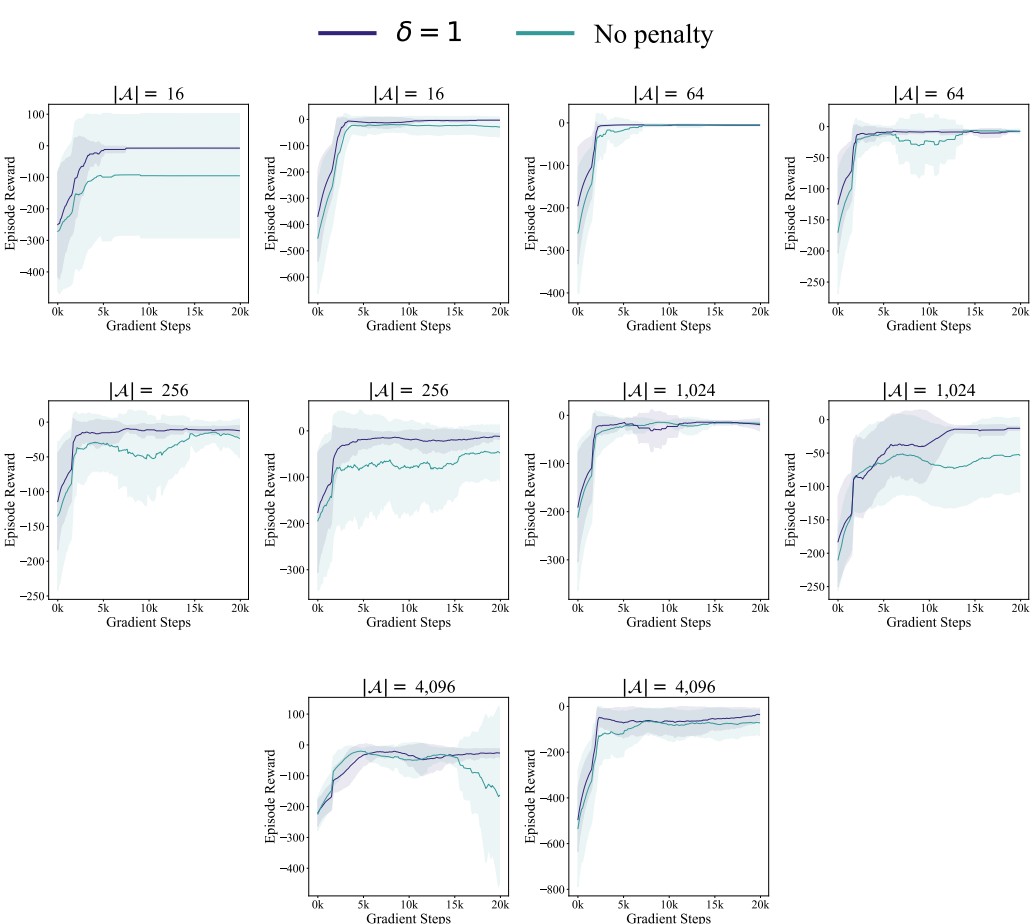

Figure 10: Learning curves for BVE agents with and without a depth penalty $\delta$ show that the former learns more quickly and achieves superior asymptotic performance than the latter.

## B.2 VARYING TD WEIGHT IN LOSS

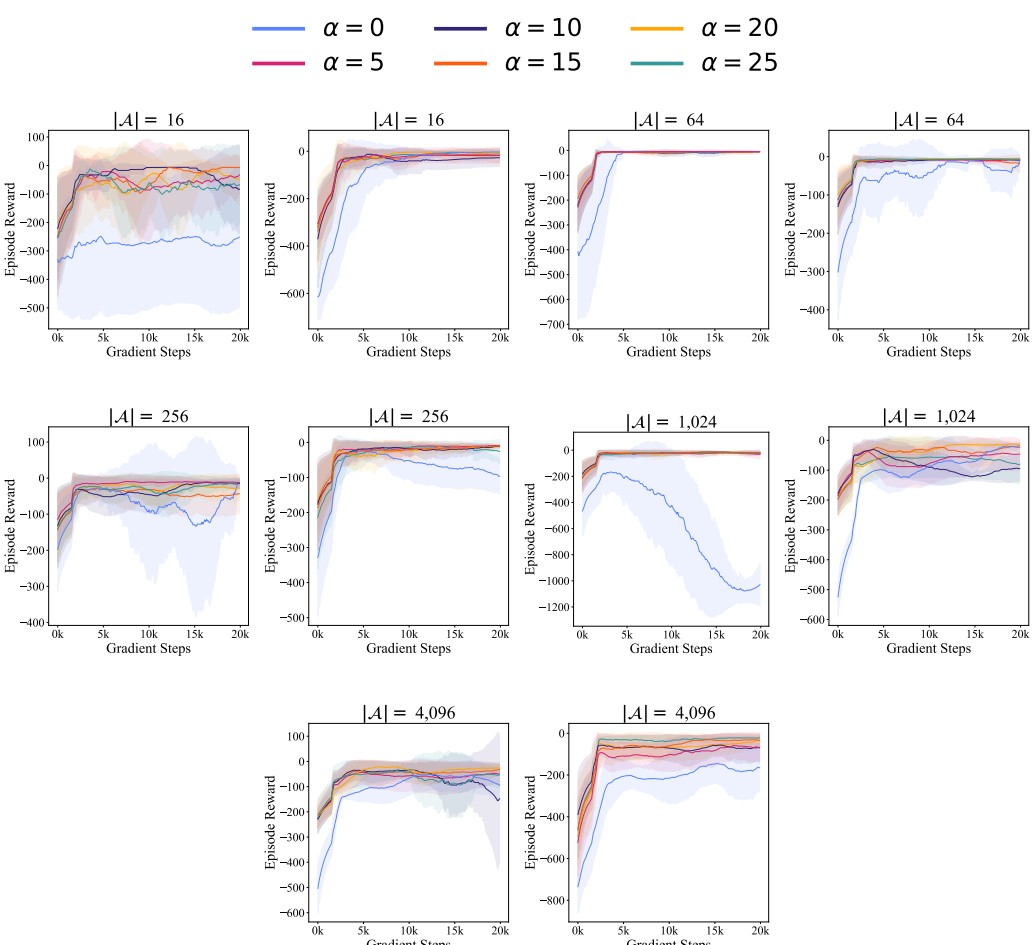

Figure 11: Learning curves for BVE agents with varying $\alpha$ values indicate stable performance, particularly in low-dimensional environments where even $\alpha = 0$ is sometimes effective. In high-dimensional settings, larger $\alpha$ values tend to improve results.

## B.3 COMPARISON TO DQN

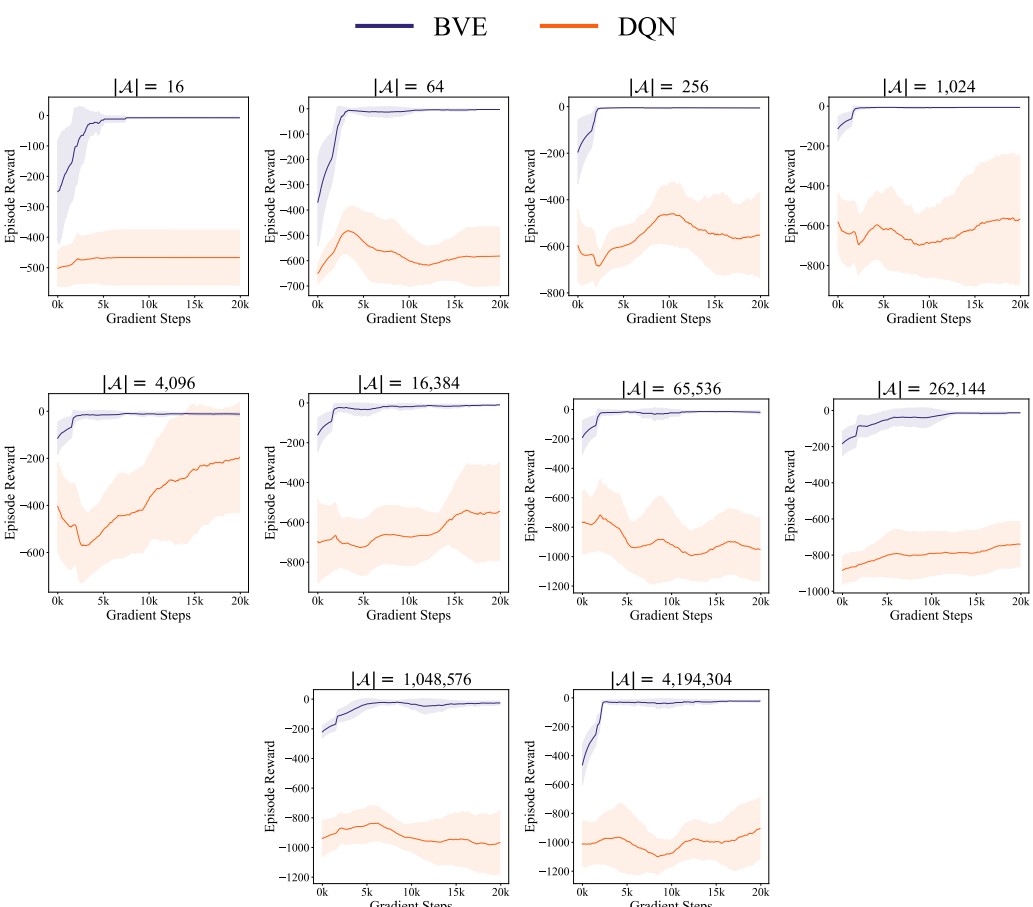

Figure 12: Despite being trained with the same behavior-regularized TD loss function as BVE and constrained to actions in $\mathcal{B}$, learning curves show that the DQN fails to learn an effective policy.

