# OpenReview forum: "Offline Reinforcement Learning With Combinatorial Action Spaces"
_ICLR.cc/2025/Conference — Submitted to ICLR 2025_

### Official Review · Reviewer_8i81 · 2024-10-30

**Soundness:** 3
**Presentation:** 2
**Contribution:** 2
**Rating:** 3
**Confidence:** 3

**Summary:**

This paper addresses the challenges of reinforcement learning in combinatorial action spaces, which arise from executing multiple sub-actions simultaneously. The exponential growth in action space size and the interdependencies among sub-actions complicate learning, particularly in offline settings with limited and suboptimal data. Current methods often simplify this by assuming independence among sub-actions. The authors propose Branch Value Estimation (BVE), a method that effectively captures these dependencies and scales to large combinatorial spaces by evaluating only a small subset of actions at each timestep. Experimental results demonstrate that BVE outperforms baseline approaches across various action space sizes.

**Strengths:**

The motivation of this paper is clear: the issue of combinatorial action spaces in offline reinforcement learning has indeed been underexplored, making it worthy of further attention. The experiments presented are also quite persuasive and effectively demonstrate the advantages of BVE.

**Weaknesses:**

1. The method introduction in this paper lacks clarity, particularly regarding the process of constructing the tree and the Q-learning based on the tree, making it difficult to understand the underlying design principles. For example, in the tree shown in Figure 1, are the elements in the third and fourth layers fixed, or can their positions be swapped? Additionally, the output of \( f(s,a) \) includes \( v \), but based on the example in Figure 3, the dimensions of \( v \) corresponding to different layers of the tree are different. So what is the dimensionality of the output of \( f(s,a) \)? Is it fixed or variable? I haven't found definitive answers to these questions in the paper.

2. The paper claims that the proposed method can eliminate the assumption of action space independence from previous works; however, it does not explicitly explain why BVE can effectively consider the interdependencies among different sub-actions, merely presenting the process without clearly outlining its advantages.  So I recommend the author to provide a more explicit explanation or analysis of how BVE captures sub-action dependencies.

3. I understand that the design concept may be similar to using a binary search method to find the maximum value, thereby obtaining the maximum among exponentially many action combinations with logarithmic operations. I'm not sure if my understanding is accurate, so I suggest the authors further develop the convergence theory of Q-learning under this search paradigm, which would enhance the paper.

4. Although the paper considers the problem within an offline RL framework, it only uses an optimization objective with a behavior cloning regularization term without further addressing out-of-distribution (OOD) or distribution shift issues. This makes the offline RL setting seem unnecessary here, so the authors might consider examining the method's performance in an online setting. Additionally, I recommend the authors refer to the following paper, which, while focused on multi-agent reinforcement learning, also addresses the OOD problem in combinatorial action spaces in offline settings using a counterfactual Q-learning design, which could be relevant to this paper's problem setup.

5. Regarding the experimental section, I find the current experimental setup somewhat artificial and lacking in realistic tasks. I suggest the authors conduct experiments in more meaningful real-world scenarios.

[1] Counterfactual Conservative Q Learning for Offline Multi-agent Reinforcement Learning

**Questions:**

The same as the questions in Weaknesses.

---

> ### Author Response · Authors · 2024-11-25
> **Response to Reviewer 8i81 Part 1**
>
> We would like to thank the reviewer for their positive comments and constructive feedback. We appreciate your recognition of the importance of the problem setting and the persuasiveness of our results. In response to your concerns, we have conducted additional experiments and have addressed all points raised.
>
> We believe our responses and additional experiments sufficiently address your concerns and provide clarity. We would greatly appreciate it if you could reassess your evaluation based on these updates. Should any concerns remain, we welcome further feedback on how we might refine our work to align with your expectations for a higher score. Thank you again for your thoughtful review.
>
> ---
>
> ### W1.1 Tree Construction: In the tree shown in Figure 1, are the elements in the third and fourth layers fixed, or can their positions be swapped?
>
> While the nodes in the tree are not inherently ordered, once an order is established, it becomes fixed. For instance, in a 2D grid world, the tree can be constructed such that the left-most node corresponds to the "up," "down," "left," or "right" direction. However, once the tree is constructed, the sub-action assigned to the left-most node remains constant — it cannot represent "up" in one iteration and "down" in another. This principle applies uniformly to every node across all layers of the tree. This has been clarified in Section 3.
>
>
> ### W1.2 Tree Construction: The output of ( f(s,a) ) includes ( v ), but based on the example in Figure 3, the dimensions of ( v ) corresponding to different layers of the tree are different. So what is the dimensionality of the output of ( f(s,a) )? Is it fixed or variable?
>
> We appreciate the opportunity to clarify the dimensionality of our network's output. As the reviewer notes, the number of children for each node varies. The network is designed to output one more than the maximum possible number of children in the tree, with the additional output used to represent the Q-value of the current node. For example, in Figure 1, the network produces five outputs, as the nodes [0,0,0], [1,0,0], and [2,0,0] have four children. The number of network outputs remains fixed. For nodes with fewer than the maximum number of children, masking is applied to eliminate invalid outputs. This approach aligns with established techniques in both policy gradient algorithms [1] and value-based methods, such as AlphaZero [2].
>
> We have added more detail about the output dimensionality in Section 3.

---

> ### Author Response · Authors · 2024-11-25
> **Response to Reviewer 8i81 Part 2**
>
> ### W2 Accounting for Sub-Action Dependence: I recommend the author to provide a more explicit explanation or analysis of how BVE captures sub-action dependencies
>
> We note that the standard Q-learning loss inherently accounts for sub-action dependencies, as the Q-value is defined over the complete action. However, in combinatorial discrete spaces, estimating Q-values for entire actions is challenging due to the exponential growth in the number of possible actions with the number of sub-actions. To address this, factored approaches approximate Q-values by aggregating contributions from individual components, such as through linear decomposition. While effective under certain conditions, these approaches are limited when sub-action dependencies are strong or reward structures are not factorizable. These issues are well-documented in [3] (Sections 4.1 and 4.2) and [4] (Section 3.1) and apply to all factorized methods, regardless of the underlying algorithm. BVE, in contrast, estimates Q-values for entire actions, inherently capturing sub-action dependencies. By evaluating only a small subset of possible actions at each timestep, it mitigates the curse of dimensionality that standard Q-learning faces.
>
> Our experiments in Section 4.1 demonstrate that BVE is robust to diverse reward structures, while those in Section 4.2 confirm that BVE explicitly accounts for sub-action dependencies. To further evaluate BVE's ability to account for sub-action dependencies, we conducted a new set of experiments in environments with varying levels of sub-action dependence. This was achieved by increasing the number of pits in the environment, as a higher pit density demands greater coordination between sub-actions. To ensure a viable path from the start state to the goal state, pits were restricted to interior (non-boundary) states, which total $3^N$ in the grid world. Specifically, we generated six new datasets in an 8D grid world with 328, 656, 1640, 3281, 4921, and 6561 pits, corresponding to 5%, 10%, 25%, 50%, 75%, and 100% of interior states being pits.
>
> The 8D grid world was chosen as it strikes a balance between action-space complexity (65,536 possible actions per state) and the computational feasibility of creating multiple new datasets.
>
> | Number of pits | BVE                       | FAS                | IQL               |
> |----------------|---------------------------|--------------------|-------------------|
> | 0              | $\mathbf{-8.2 \pm 2.2}$   | $-24.4 \pm 10.4$   | $-28.1 \pm 12.1$  |
> | 328            | $\mathbf{-19.0 \pm 2.3}$  | $-78.4 \pm 99.7$.  | $-85.8 \pm 45.0$  |
> | 656            | $\mathbf{-16.8 \pm 3.3}$  | $-140.4 \pm 177.1$ | $-88.1 \pm 50.1$  |
> | 1640           | $\mathbf{-42.9 \pm 43.3}$ | $-178.6 \pm 212.8$ | $-82.8 \pm 52.0$  |
> | 3281           | $\mathbf{-54.8 \pm 47.9}$ | $-902.6 \pm 274.1$ | $-106.4 \pm 42.8$ |
> | 4921           | $\mathbf{-42.9 \pm 34.3}$ | $-942.8 \pm 278.1$ | $-110.0 \pm 34.3$ |
> | 6561           | $\mathbf{-41.6 \pm 22.4}$ | $-1131.4 \pm 0.0$  | $-112.4 \pm 22.8$ |
>
>
> As shown in the table above, BVE's performance remains stable across all environments, whereas FAS's performance declines significantly when half the interior states are pits. Similarly, IQL's performance deteriorates as the number of pits — and thus the dependencies among sub-actions — increases. Notably, in the environment with 6,561 pits, all interior states are pits, requiring precise coordination of every sub-action to avoid catastrophic failure; a single sub-action misstep into an interior state results in the agent falling into a pit. This experiment has been added to Section 4.
>
>
> ### W3 Tree Search: I understand that the design concept may be similar to using a binary search method to find the maximum value, thereby obtaining the maximum among exponentially many action combinations with logarithmic operations. I suggest the authors further develop the convergence theory of Q-learning under this search paradigm
>
> The reviewer's intuition about the tree search paradigm and its utility in combinatorial action spaces is accurate. By structuring the action space as a tree, BVE requires predictions for only a small subset of Q-values at each timestep. Regarding convergence theory, BVE inherits the convergence guarantees of tabular Q-learning in the tabular case [5]. While we agree that a rigorous theoretical analysis of BVE's convergence properties under function approximation would be valuable, this falls outside the scope of the current work. We view this as an important avenue for future research, building upon the foundations established in this study.

---

> ### Author Response · Authors · 2024-11-26
> **Response to Reviewer 8i81 Part 3**
>
> ### W4.1 Application to Offline RL: The paper only uses an optimization objective with a behavior cloning regularization term without further addressing OOD or distribution shift issues. This makes the offline RL setting seem unnecessary here
>
> We appreciate the opportunity to clarify how BVE leverages established offline RL principles to mitigate overestimation bias.
>
> The first mechanism is the behavior cloning term $\|\hat{\mathbf{a}}' - \mathbf{a}'\|$ in Equation 4, which encourages the agent to select actions similar to those of the behavior policy. This term can be viewed as a value-based analog to the behavior cloning regularizer introduced in TD3+BC (Equation 3). Despite its simplicity, such regularization has been shown to be effective across diverse problem settings [6].
>
> Second, we sparsify the action tree to restrict Q-value evaluations to actions present in the dataset. Restricting evaluations to dataset actions was first proposed in [7] (BCQ) and is further detailed in [8]. While BCQ uses a generative model to achieve this, BVE accomplishes it by sparsifying the action tree.
>
> In response to the reviewer’s suggestion, we have further elaborated on these mechanisms in Section 3.
>
>
> ### W4.2 Application to Offline RL: The authors might consider examining the method's performance in an online setting
>
> We agree that BVE is well-suited for online RL. However, our focus on offline RL is motivated by several key considerations. First, offline RL is crucial for real-world applications, many of which have combinatorial action spaces — examples include healthcare systems, traffic light optimization, and large-scale industrial processes. In these domains, online execution of a learning algorithm is infeasible and likely unsafe. Additionally, in many cases, there is a breadth of historical data that can be used for model training. Despite the significance and prevalence of these applications, this problem setting has received comparatively limited attention in the literature.
>
> Second, as described in W4.1, BVE's sparsified tree structure mitigates overestimation error by constraining evaluation to actions observed in the training dataset. While BCQ [7] uses a similar principle, it relies on modeling the behavior policy through a conditional VAE — an approach that presents challenges for online fine-tuning [9]. BVE offers a more elegant solution: its sparse tree naturally restricts the action space in combinatorial settings without requiring explicit behavior policy modeling, thereby enabling seamless online fine-tuning after offline RL.
>
> To validate BVE's online fine-tuning capability, we compared it with IQL, which is known for strong performance in online fine-tuning after offline RL. (FAS was excluded as, being a factorized variant of BCQ, it is unsuitable for online fine-tuning, as previously noted.) As discussed in W2, we conducted additional experiments to evaluate BVE's robustness across environments with varying sub-action dependencies, controlled by the number of pits. Relevant to our fine-tuning experiments is that, while BVE outperforms baselines in all environments, it faces its greatest challenge in the environment with 3,281 pits. This environment is particularly challenging because the agent must avoid actions in states near the starting state that become optimal only after the agent navigates past the pit. We therefore selected this most demanding case for fine-tuning, as success here implies generalizability to environments where BVE performs better.
>
> After 1,500 gradient steps of offline training — at which point BVE achieves a reward of -226.1 compared to IQL's -259.4 — we fine-tuned both policies for 5,000 timesteps. Offline training was stopped after 1,500 gradient steps as, beyond this point, BVE's performance began to surpass that of IQL. Notably, BVE discovers a policy that not only exceeds the performance of its offline counterpart but also slightly outperforms IQL's online policy.
>
> | Setting | BVE                       | IQL               |
> |---------|---------------------------|-------------------|
> | Offline | $\mathbf{-54.8 \pm 47.9}$ | $-106.4 \pm 42.8$ |
> | Online  | $\mathbf{-24.5 \pm 2.6}$  | $-28.6 \pm 13.1$  |
>
> This experiment is now included in Section 4 of our paper.
>
>
> ### W4.3 Application to Offline RL: I recommend the authors refer to the following paper... which could be relevant to this paper's problem setup
>
> We thank the reviewer for highlighting this relevant work. It has been included in our discussion of related work in Section 5.

---

> ### Author Response · Authors · 2024-11-26
> **Response to Reviewer 8i81 Part 4**
>
> ### W5 Experimental Setup: I suggest the authors conduct experiments in more meaningful real-world scenarios.
>
> We appreciate the reviewer’s suggestion. The primary objective of our experimental setup is to demonstrate BVE's ability to learn effectively in environments with strong sub-action dependencies. We have addressed this in several ways. As outlined in W2, we show that BVE performs well when reward functions are non-factorizable and when sub-action dependencies vary. Furthermore, the inability of state-of-the-art baselines to learn performant policies in many of our testing environments highlights the challenges inherent in these settings, reinforcing the validity and rigor of our experimental design.

---

> ### Author Response · Authors · 2024-11-26
> **References Cited in Response to Reviewer 8i81**
>
> [1] Shengyi Huang and Santiago Ontañón. 2020. A closer look at invalid action masking in policy gradient algorithms. arXiv preprint arXiv:2006.14171 (2020).
>
> [2] David Silver, Thomas Hubert, Julian Schrittwieser, Ioannis Antonoglou, Matthew Lai, Arthur Guez, Marc Lanctot, Laurent Sifre, Dharshan Kumaran, Thore Graepel, and others. 2018. A general reinforcement learning algorithm that masters chess, shogi, and Go through self-play. Science 362, 6419 (2018), 1140–1144.
>
> [3] Alex Beeson, David Ireland, and Giovanni Montana. 2024. An Investigation of Offline Reinforcement Learning in Factorisable Action Spaces. Transactions on Machine Learning Research.
>
> [4] Shengpu Tang, Maggie Makar, Michael W. Sjoding, Finale Doshi-Velez, and Jenna Wiens. 2023. Leveraging Factored Action Spaces for Efficient Offline Reinforcement Learning in Healthcare (2023).
>
> [5] Richard S. Sutton and Andrew G. Barto. 2018. Reinforcement Learning: An Introduction. A Bradford Book, Cambridge, MA, USA.
>
> [6] Scott Fujimoto and Shixiang Shane Gu. 2021. A minimalist approach to offline reinforcement learning. Advances in neural information processing systems 34, (2021), 20132–20145.
>
> [7] Scott Fujimoto, David Meger, and Doina Precup. 2019. Off-policy deep reinforcement learning without exploration. In International conference on machine learning, PMLR, 2052–2062.
>
> [8] Sergey Levine, Aviral Kumar, George Tucker, and Justin Fu. 2020. Offline reinforcement learning: Tutorial, review, and perspectives on open problems. arXiv preprint arXiv:2005.01643 (2020).
>
> [9] Ashvin Nair, Abhishek Gupta, Murtaza Dalal, and Sergey Levine. 2020. Awac: Accelerating online reinforcement learning with offline datasets. arXiv preprint arXiv:2006.09359 (2020).

---

> ### Author Response · Authors · 2024-12-01
> **Follow up on rebuttal**
>
> Thank you for your feedback and for helping us improve our work! We would like to kindly remind you that we have resolved your concerns. As the discussion period concludes, please do not hesitate to let us know if there’s anything else we can address before it closes.

---

### Official Review · Reviewer_oUY2 · 2024-11-02

**Soundness:** 2
**Presentation:** 3
**Contribution:** 2
**Rating:** 3
**Confidence:** 3

**Summary:**

The paper addresses the challenge of combinatorial action spaces in offline reinforcement learning by proposing a method called Branch Value Estimation (BVE). BVE utilizes a tree structure to effectively prune the action space, organizing actions into a hierarchy where each node in the tree represents a sub-action conditioned on the values from its parent node. This structure, to some extent, captures dependencies among sub-actions and also reduces the number of actions evaluated at each timestep. As the tree is traversed, BVE estimates the highest achievable Q-value at each branch, allowing for the selection of the optimal action. The effectiveness of BVE is demonstrated through comparisons with state-of-the-art offline RL algorithms across various action space sizes, showing improved performance in environments with up to over 4 million possible actions.

**Strengths:**

1. The paper explains its main idea clear.
2. The paper targets an important problem.
3. The application of tree-structured action search seems new.

**Weaknesses:**

The critique identifies several weaknesses that should be addressed:

Clarity on Action Space Reduction: The paper states that BVE reduces the effective action space by organizing it into a tree structure, unlike traditional RL methods which often misidentify the optimal next action a^′\hat{a}’a^′ in equation 4. However, it remains unclear whether the overestimated actions are effectively excluded by this method. The tree traversal procedure appears optimistic about the estimated action values. It is not evident from the paper which design choice specifically helps to prevent overestimation in an offline setting, as the behavior cloning itself contributes to this effect. If the authors want to claim the tree-based method can mitigate overestimation, either theoretical or empirical evidence should be provided.

Application to Online RL: Given that the tree structure concept for managing large action spaces could also be beneficial in online RL settings—which are generally less challenging than offline settings—it’s surprising that the authors chose not to begin with online RL.

Broader Applicability and Contributions: The potential for applying the tree-structure approach in other offline RL methods raises questions about the broader significance of the work. If applicable, the contribution could be more substantial.

Ablation Study and Its Implications: The message of the ablation study is ambiguous. Figure 1 only displays two deltas, with Figure (b) showing a difference only when alpha=0, while Figure (c) suggests that DQN performs poorly even in small action spaces, raising questions about its suitability as a competitor. Additionally, it’s unclear whether the DQN is used in an offline learning setting, which would be unusual.

Justification for Design Choices: Equation (4) appears abruptly, with no justification for discounting ||a' - \hat{a}‘||. The paper should clarify why this design was chosen.

Addressing Dual Challenges: The authors attempt to address both the challenges of large discrete action spaces and offline learning. This dual focus can cause confusion regarding the contributions and effects of the design choices. Two primary questions should be addressed:
Scalability: As the action space increases, can this approach maintain high computational and sample efficiency compared to other baselines? This should be the first point to verify given that the proposed idea mainly addresses scalability. Learning curves in the form of performance v.s. computation time/number of samples are expected. Overestimation in Offline Settings: How does this approach avoid the overestimation problem in offline settings, or what offline RL algorithms can be integrated if it does not?

Empirical Results and Parameter Selection: The empirical results in Figures 8 and 9 do not show statistically significant differences, raising concerns about the effectiveness. The selection process for IQL and FAS hyperparameters is also not clearly described.

Omission of Related Work: The paper fails to cite several highly relevant works, including:
“Conjugate Markov Decision Processes” by Philip Thomas et al., which deals with extremely large action spaces.
“Reinforcement Learning with Function-Valued Action Spaces for Partial Differential Equation Control” by Yangchen Pan et al., applicable to high-dimensional continuous control and potentially adaptable for discrete settings.
“Deep Reinforcement Learning in Large Discrete Action Spaces” by Gabriel Dulac-Arnold et al.
“Conditionally optimistic exploration for cooperative deep multi-agent reinforcement learning” by Xutong Zhao et al., which utilizes a tree-structured action for exploration purposes in a multi-agent, centralized setting. Note that, although it is a multi-agent setting, the subagents' actions are concatenated to a vector for execution, which can be thought of as one agent with high dimensional discrete actions.

**Questions:**

see above.

---

> ### Author Response · Authors · 2024-11-25
> **Response to Reviewer oUY2 Part 1**
>
> Thank you for recognizing the novelty and practical significance of our approach to addressing an important yet under-explored problem. We have carefully considered your feedback, conducted additional experiments, and performed further analyses based on your suggestions.
>
> We believe our responses and the new experiments adequately address your concerns, and we kindly request your reconsideration of our score. If any issues remain, we would greatly appreciate further guidance on how we might improve our work to meet the standards for a higher evaluation. Thank you again for your time and feedback.
>
> ---
>
> ### W1 Clarity on Action Space Reduction: It is not evident from the paper which design choice specifically helps to prevent overestimation in an offline setting
>
> We appreciate the opportunity to clarify how BVE draws from well-established offline RL principles to reduce overestimation bias.
>
> The first mechanism is the behavior cloning term $\|\hat{\mathbf{a}}' - \mathbf{a}'\|$ in Equation 4, which encourages the agent to select actions similar to those chosen by the behavior policy. This term can be viewed as a value-based analog to the behavior cloning regularizer introduced in TD3+BC (Equation 3). Despite its simplicity, such regularization has been shown to be effective across diverse problem settings [1].
>
> Second, we sparsify the action tree to restrict Q-value evaluations to actions present in the dataset. Restricting evaluations to dataset actions was first proposed in [2] (BCQ) and is further detailed in [3]. While BCQ uses a generative model to achieve this, BVE accomplishes it by sparsifying the action tree.
>
> As per the reviewer’s suggestion, we have elaborated on these mechanisms in Section 3.

---

> > ### Author Response · Authors · 2024-11-25
> > **Response to Reviewer oUY2 Part 4**
> >
> > ### W5 Justification for Design Choices: Equation (4) appears abruptly, with no justification for discounting ||a' - \hat{a}‘||. The paper should clarify why this design was chosen.
> >
> > We thank the reviewer for the insightful feedback on Equation 4. We agree that the paper should explicitly clarify the rationale for discounting the behavior cloning regularizer $\|\hat{\mathbf{a}}' - \mathbf{a}' \|$. We included this term within the discount factor $\gamma$ to treat the imitation penalty as a future cost, aligning it with the temporal structure of RL. By framing deviations from the expert policy as part of the expected future returns, we found that this approach stabilizes learning. It encourages the agent to prioritize long-term adherence to the behavior policy without disproportionately affecting the immediate reward signal. We have revised the manuscript to clearly articulate this design choice and provide its justification.
> >
> >
> > ### W6.1 Addressing Dual Challenges: As the action space increases, can this approach maintain high computational and sample efficiency compared to other baselines?
> >
> > Figures 8 and 9 demonstrate that BVE's sample efficiency consistently matches or exceeds that of the baselines across nearly all environments. Specifically, in low-dimensional environments without pits, all methods achieve policy stabilization after a comparable number of gradient steps. In higher-dimensional environments without pits, BVE outperforms both baselines: IQL requires significantly more gradient steps, while FAS becomes unstable as the discrepancy between the linearly decomposed and true reward functions grows. In the three lowest-dimensional environments with pits, IQL converges with the fewest gradient steps. However, in all higher-dimensional environments, BVE demonstrates superior efficiency, requiring fewer steps than both baselines.
> >
> > In addition, we emphasize that in most environments, BVE learns a viable policy while state-of-the-art baselines fail. A key condition for BVE's effectiveness is a combinatorial, discrete action space, which aligns naturally with its tree structure. However, BVE's true strength emerges in environments where sub-actions are strongly dependent or reward functions are not factorizable. In such cases, as demonstrated in Section 4 and supported theoretically [5, 6], current state-of-the-art methods are less effective, regardless of the number of gradient steps taken.
> >
> >
> > ### W6.2 Addressing Dual Challenges: How does this approach avoid the overestimation problem in offline settings, or what offline RL algorithms can be integrated if it does not?
> >
> > As outlined in W1, we use two mechanisms to mitigate overestimation error: a behavior cloning term and sparsification of the action space tree. These strategies are adapted from well-established offline RL methods. Furthermore, as discussed in W3, the tree structure is broadly applicable and not inherently tied to Equation 4. A thorough analytical analysis of BVE, including the integration of alternative offline RL algorithms, remains an important direction for future work.

---

> ### Author Response · Authors · 2024-11-25
> **Response to Reviewer oUY2 Part 2**
>
> ### W2 Application to Online RL: Given that the tree structure concept for managing large action spaces could also be beneficial in online RL settings — which are generally less challenging than offline settings —it’s surprising that the authors chose not to begin with online RL
>
> We agree that BVE is well-suited for online RL. However, we focused on offline RL for several reasons. Offline RL is critical for real-world applications, many of which involve combinatorial action spaces—examples include healthcare applications, traffic light control, and large industrial processes. In these domains, online execution of a learning algorithm is infeasible and likely unsafe. Additionally, in many cases, there is a breadth of historical data that can be used for model training. Despite the importance and ubiquity of these applications, this problem setting has received relatively limited attention.
>
> Second, as described in W1, BVE's sparsified tree structure mitigates overestimation error by constraining evaluation to actions observed in the dataset. While BCQ [2] uses a similar principle, it relies on modeling the behavior policy through a conditional VAE — an approach that presents challenges for online fine-tuning [4]. BVE offers a more elegant solution: its sparse tree naturally restricts the action space in combinatorial settings without requiring explicit behavior policy modeling, thereby enabling seamless online fine-tuning after offline RL.
>
> To validate BVE's online fine-tuning capability, we compare it with IQL, noted for its similar strength in online fine-tuning after offline RL (FAS is excluded from this experiment since, as a factorized version of BCQ, it is not suited to online fine-tuning). As outlined in W4.1 of our response to reviewer ahEi, we conducted additional experiments to evaluate BVE's robustness across environments with varying sub-action dependencies, controlled by the number of pits. Relevant to our fine-tuning experiments is that, while BVE outperforms baselines in all environments, it faces its greatest challenge in the environment with 3,281 pits. This environment is particularly challenging because the agent must avoid actions in states near the starting state that become optimal only after the agent navigates past the pit. We therefore selected this most demanding case for fine-tuning, as success here implies generalizability to environments where BVE performs better.
>
> After 1,500 gradient steps of offline training — at which point BVE achieves a reward of -226.1 compared to IQL's -259.4 — we fine-tuned the policies for 5,000 timesteps. We stopped offline training after 1,500 gradient steps because, after this point, BVE's performance began to surpass IQL's. Importantly, BVE finds a policy that not only surpasses the one learned offline but also slightly outperforms IQL's online policy.
>
> | Setting | BVE                       | IQL               |
> |---------|---------------------------|-------------------|
> | Offline | $\mathbf{-54.8 \pm 47.9}$ | $-106.4 \pm 42.8$ |
> | Online  | $\mathbf{-24.5 \pm 2.6}$  | $-28.6 \pm 13.1$  |
>
> This experiment has been included in Section 4 of our paper.
>
>
> ### W3 Broader Applicability and Contributions: The potential for applying the tree-structure approach in other offline RL methods raises questions about the broader significance of the work. If applicable, the contribution could be more substantial.
>
> We appreciate the reviewer highlighting the significance of BVE's tree structure. Indeed, we agree that the advantages of this approach extend beyond our specific implementation. While the loss function in Equation 4 is both simple and effective, the tree structure itself is not inherently tied to this particular function. Moreover, as the reviewer noted in W2, the utility of the tree structure is not confined to the specific setting explored in our work. We believe this approach holds promise in various combinatorial contexts, with constrained RL being a particularly compelling example.
>
> We feel this work provides a foundation for future research on combinatorial action spaces. To avoid overly broad claims, however, we focused specifically on offline RL, employing the loss function defined in Equation 4.

---

> ### Author Response · Authors · 2024-11-25
> **Response to Reviewer oUY2 Part 3**
>
> ### W4.1 Ablation Study and Its Implications: Figure 1 only displays two deltas
>
> Per the reviewer's suggestion, we extended our original ablation study — previously limited to $\delta = 1$ and no depth penalty — to evaluate $\delta$ values of 5, 10, and 15:
>
> | $\mathcal{A}$ | No penalty               | $\delta = 1$              | $\delta = 5$       | $\delta = 10$      | $\delta = 15$      |
> |---------------|--------------------------|---------------------------|--------------------|--------------------|--------------------|
> | 16            | $-95.2 \pm 197.1$        | $\mathbf{-7.5 \pm 2.4}$   | $-124.3 \pm 168.7$ | $-97.2 \pm 91.7$   | $-83.3 \pm 86.2$   |
> | 64            | $-28.8 \pm 37.9$         | $\mathbf{-2.9 \pm 0.2}$   | $-45.0 \pm 37.0$   | $-70.3 \pm 9.5$    | $-70.6 \pm 62.6$   |
> | 256           | $\mathbf{-4.5 \pm 0.4}$  | $-5.6 \pm 1.5$            | $-64.7 \pm 37.2$   | $-38.4 \pm 42.9$   | $-81.8 \pm 54.2$   |
> | 1024          | $\mathbf{-7.6 \pm 3.8}$  | $-7.7 \pm 1.4$.           | $-48.1 \pm 49.5$   | $-474.8 \pm 462.6$ | $-416.0 \pm 426.9$ |
> | 4096          | $-23.5 \pm 28.3$         | $\mathbf{-12.4 \pm 7.9}$  | $-29.2 \pm 37.9$   | $-320.0 \pm 470.0$ | $-811.0 \pm 331.2$ |
> | $\sim$16k     | $-47.4 \pm 61.3$         | $\mathbf{-11.9 \pm 6.9}$  | $-61.9 \pm 56.2$   | $-89.6 \pm 127.2$  | $-545.4 \pm 518.9$ |
> | $\sim$65k     | $\mathbf{-16.6 \pm 4.7}$ | $-19.4 \pm 13.2$          | $-371.5 \pm 512.0$ | $-875.5 \pm 348.1$ | $-1116.8 \pm 30.0$ |
> | $\sim$260k    | $-53.3 \pm 56.0$         | $\mathbf{-12.9 \pm 3.0}$  | $-110.4 \pm 37.7$  | $-89.7 \pm 60.1$   | $-76.3 \pm 62.6$   |
> | $\sim$1M      | $-163.5 \pm 281.8$       | $\mathbf{-26.3 \pm 14.1}$ | $-78.3 \pm 58.9$   | $-82.2 \pm 64.6$   | $-187.6 \pm 132.4$ |
> | $\sim$4M      | $-71.3 \pm 56.2$         | $\mathbf{-35.9 \pm 28.8}$ | $-123.1 \pm 87.6$  | $-165.2 \pm 150.5$ | $-270.3 \pm 221.6$ |
>
> The results indicate that while a depth penalty generally improves performance, it should remain small, with $\delta = 1$ yielding the best results across nearly all environments. This experiment has been added to Section 4 of our paper.
>
>
> ### W4.2 Ablation Study and Its Implications: Figure (b) showing a difference only when alpha=0
>
> We appreciate the opportunity to clarify the impact of $\alpha$ on policy performance, as we agree that Figure 7(b) could be improved to better address outlier values (e.g., $\alpha = 0$ for the environment with approximately 65k actions). In general, we observe that in low-dimensional settings, $\alpha$ has minimal influence on performance. However, in higher-dimensional environments, larger $\alpha$ values tend to yield superior results. To illustrate this, we can compare the results for the environment with 256 actions:
>
> | Alpha | Return         |
> |-------|----------------|
> | 0     | $-4.5 \pm 0.4$ |
> | 5     | $-4.9 \pm 0.8$ |
> | 10    | $-4.6 \pm 0.4$ |
> | 15    | $-5.6 \pm 2.8$ |
> | 20    | $-4.6 \pm 0.5$ |
> | 25    | $-5.1 \pm 1.6$ |
>
> To those for the environment with over 4 million actions:
>
> | Alpha | Return             |
> |-------|--------------------|
> | 0     | $-164.8 \pm 118.3$ |
> | 5     | $-70.3 \pm 73.0$   |
> | 10    | $-68.0 \pm 65.0$   |
> | 15    | $-34.3 \pm 27.5$   |
> | 20    | $-35.9 \pm 28.8$   |
> | 25    | $-21.4 \pm 4.7$    |
>
> We have adjusted the y-axis scale in Figure 8(b) to exclude outliers, providing a clearer illustration of this difference.
>
> ### W4.3 Figure (c) suggests that DQN performs poorly even in small action spaces, raising questions about its suitability as a competitor
>
> This experiment was designed to highlight the necessity of BVE's tree structure. Notably, we did not train a standard DQN; instead, it was trained using the loss in Equation 4 and restricted to select only actions within the dataset — two mechanisms BVE relies on to enhance offline learning, as detailed in W1. Effectively, this DQN variant represents BVE without the tree search and loss propagation described in Section 3. Crucially, this DQN variant was not intended to serve as a direct competitor to BVE, which is why it was included as an ablation study rather than as a baseline in our primary experiments.

---

> ### Author Response · Authors · 2024-11-25
> **Response to Reviewer oUY2 Part 5**
>
> ### W7.1 Empirical Results and Parameter Selection: The empirical results in Figures 8 and 9 do not show statistically significant differences, raising concerns about the effectiveness.
>
> Figures 8 and 9 present the full learning curves corresponding to the results summarized in Figures 5 and 6, respectively. In low-dimensional environments without pits, FAS performs comparably to BVE. However, in higher-dimensional environments without pits, the discrepancy between the linearly decomposed and true reward functions becomes more pronounced, resulting in instability in FAS's learning and a decline in performance, while BVE consistently outperforms IQL. In environments with pits, BVE demonstrates clear superiority over both FAS and IQL across all settings.
>
> We conducted a new set of experiments to further validate these results. Specifically, we designed environments with varying levels of dependence among sub-actions by increasing the number of pits — greater coordination between sub-actions is required as the number of pits increases. To ensure a viable path from the start to the goal state, pits were restricted to interior (non-boundary) states, which total $3^N$ in the grid world. We generated six new datasets in the 8D grid world, with 328, 656, 1640, 3281, 4921, and 6561 pits, corresponding to 5%, 10%, 25%, 50%, 75%, and 100% of interior states, respectively, being pits.
>
> We selected the 8D world as it balances action-space complexity (65,536 possible actions per state) with the computational feasibility of generating multiple datasets.
>
> | Number of pits | BVE                       | FAS                | IQL               |
> |----------------|---------------------------|--------------------|-------------------|
> | 0              | $\mathbf{-8.2 \pm 2.2}$   | $-24.4 \pm 10.4$   | $-28.1 \pm 12.1$  |
> | 328            | $\mathbf{-19.0 \pm 2.3}$  | $-78.4 \pm 99.7$.  | $-85.8 \pm 45.0$  |
> | 656            | $\mathbf{-16.8 \pm 3.3}$  | $-140.4 \pm 177.1$ | $-88.1 \pm 50.1$  |
> | 1640           | $\mathbf{-42.9 \pm 43.3}$ | $-178.6 \pm 212.8$ | $-82.8 \pm 52.0$  |
> | 3281           | $\mathbf{-54.8 \pm 47.9}$ | $-902.6 \pm 274.1$ | $-106.4 \pm 42.8$ |
> | 4921           | $\mathbf{-42.9 \pm 34.3}$ | $-942.8 \pm 278.1$ | $-110.0 \pm 34.3$ |
> | 6561           | $\mathbf{-41.6 \pm 22.4}$ | $-1131.4 \pm 0.0$  | $-112.4 \pm 22.8$ |
>
>
> As shown in the table above, BVE's performance remains relatively stable across all environments, whereas FAS's performance declines significantly when half of the interior states are pits. Similarly, IQL's performance deteriorates as the number of pits — and thus the dependencies among sub-actions — increases. This experiments has been added to Section 4.
>
>
> ### W7.2 Empirical Results and Parameter Selection: The selection process for IQL and FAS hyperparameters is also not clearly described.
>
> To ensure a fair comparison, we optimized hyperparameters for all methods. Specifically, we evaluated the number of network layers, the number of hidden nodes per layer, learning rate, learning rate decay, and batch size. Additionally, we optimized algorithm-specific hyperparameters for each method.
>
>
> ### W8 Omission of Related Work: The paper fails to cite several highly relevant works
>
> We appreciate the reviewer suggesting these related works and agree that they are relevant. We have now cited them in Section 5. However, we believe these works are not direct competitors to BVE. For instance, [7] appears to focus on high-dimensional continuous control, leaving the applicability and implications of adapting this method for combinatorial, discrete control unaddressed.
>
> Additionally, these methods are primarily designed for online learning, which is a non-trivial distinction. For example, [8] introduces an interesting approach, but COE relies on visitation counts of state-action pairs during training, which are derived from online exploration. In offline learning, where the dataset is fixed, generating accurate visitation counts is challenging. This limitation is similar to that of MCTS, as discussed in Section 5. Furthermore, COE's exploration strategy involves optimism-based bonuses to incentivize exploration of unvisited areas, a mechanism that is less applicable in offline learning, which focuses on leveraging the available dataset effectively.

---

> ### Author Response · Authors · 2024-11-25
> **References Cited in Response to Reviewer oUY2**
>
> [1] Scott Fujimoto and Shixiang Shane Gu. 2021. A minimalist approach to offline reinforcement learning. Advances in neural information processing systems 34, (2021), 20132–20145.
>
> [2] Scott Fujimoto, David Meger, and Doina Precup. 2019. Off-policy deep reinforcement learning without exploration. In International conference on machine learning, PMLR, 2052–2062.
>
> [3] Sergey Levine, Aviral Kumar, George Tucker, and Justin Fu. 2020. Offline reinforcement learning: Tutorial, review, and perspectives on open problems. arXiv preprint arXiv:2005.01643 (2020).
>
> [4] Ashvin Nair, Abhishek Gupta, Murtaza Dalal, and Sergey Levine. 2020. Awac: Accelerating online reinforcement learning with offline datasets. arXiv preprint arXiv:2006.09359 (2020).
>
> [5] Alex Beeson, David Ireland, and Giovanni Montana. 2024. An Investigation of Offline Reinforcement Learning in Factorisable Action Spaces. Transactions on Machine Learning Research.
>
> [6] Shengpu Tang, Maggie Makar, Michael W. Sjoding, Finale Doshi-Velez, and Jenna Wiens. 2023. Leveraging Factored Action Spaces for Efficient Offline Reinforcement Learning in Healthcare (2023).
>
> [7] Yangchen Pan, Amir-massoud Farahmand, Martha White, Saleh Nabi, Piyush Grover, and Daniel Nikovski. 2018. Reinforcement learning with function-valued action spaces for partial differential equation control. In International Conference on Machine Learning, PMLR, 3986–3995.
>
> [8] Xutong Zhao, Yangchen Pan, Chenjun Xiao, Sarath Chandar, and Janarthanan Rajendran. 2023. Conditionally optimistic exploration for cooperative deep multi-agent reinforcement learning. In Uncertainty in Artificial Intelligence, PMLR, 2529–2540.

---

> ### Author Response · Authors · 2024-12-01
> **Follow up on rebuttal**
>
> We truly appreciate your feedback and the opportunity to improve our work based on your input! As a reminder, we have addressed the concerns you raised. Please let us know if there’s anything else you’d like us to address before the discussion period closes.

---

### Official Review · Reviewer_ahEi · 2024-11-03

**Soundness:** 3
**Presentation:** 3
**Contribution:** 3
**Rating:** 5
**Confidence:** 5

**Summary:**

This paper tackles the problem of offline RL in combinatorial action spaces, where actions are formed by combinations of sub-actions, leading to a rapid increase in possible actions. Such action spaces challenge standard offline RL methods, particularly in handling dependencies among sub-actions and limited data availability.

The authors propose a novel method, Branch Value Estimation (BVE), to address these issues by structuring the action space as a tree to capture sub-action dependencies effectively. Their approach evaluates only a subset of possible actions, making the process more computationally efficient. BVE introduces a behavior-regularized temporal difference (TD) loss designed to reduce overestimation bias in such combinatorial settings, leading to more stable value estimates.

In their experiments, BVE consistently outperforms established baselines, including methods like Factored Action Spaces and Implicit Q-Learning, demonstrating superior performance and stability across a variety of high-dimensional environments. The results suggest that BVE is particularly effective in scenarios where standard methods fail to handle sub-action dependencies within combinatorial action spaces efficiently.

This paper’s contributions include (1) introducing the BVE method with a novel value estimation strategy for structured action spaces, (2) providing theoretical insights into the impact of sub-action dependencies on offline RL performance, and (3) an extensive evaluation demonstrating BVE’s effectiveness in complex, high-dimensional environments.

**Strengths:**

1. Originality. The paper introduces a novel way to represent combinatorial action spaces using trees. The branch value estimation technique presents a valuable solution to action selection in large spaces. The combination of beam search with RL is innovative and hadn't been applied this way before. The approach to handling sub-action dependencies through tree traversal is original. The method provides a new angle on addressing overestimation bias in offline RL.

2. Quality. The empirical results demonstrate clear performance advantages over existing approaches. The ablation studies effectively isolate the contribution of each component. The method successfully scales to large action spaces of up to 4 million actions. The consistent performance across different environment sizes shows robustness. The experimental design directly addresses the key claims.

3. Clarity. Complex concepts are explained through effective visualizations and examples. The algorithm descriptions are precise and well-detailed with clear pseudocode. The paper follows a logical progression that builds understanding. The experimental setup and results are presented transparently. The technical content strikes a good balance between depth and accessibility.

4. Significance. The paper tackles a fundamental challenge in applying RL to real-world problems with combinatorial action spaces. The computational efficiency improvements make previously intractable problems manageable. The method has potential applications in important domains like healthcare and robotics. The approach opens new directions for handling complex action spaces in RL.

5. Technical contribution. The tree-based representation effectively reduces the search space while maintaining performance. The branch value estimation technique successfully handles dependencies between sub-actions. The beam search integration provides an efficient solution for policy extraction.

**Weaknesses:**

1. Limited comparative evaluation: The paper compares against only two baselines (FAS and IQL), which leaves questions about relative performance against other approaches. Particularly relevant is the recent parallel work on factorized action spaces (https://openreview.net/forum?id=STwxyUfpNV) which offers additional baselines and testing environments.

2. Theoretical analysis: While the empirical results are promising, the paper would benefit from theoretical analysis of convergence properties and formal bounds on beam search approximation error. Understanding the conditions under which BVE is guaranteed to perform well would help practitioners apply the method with confidence. This theoretical foundation would also help characterize the relationship between approximation quality and final policy performance.

3. Technical details: The paper could better explain several important implementation aspects, particularly the interaction between depth penalty and beam search during action selection. A more detailed analysis of computational overhead would help understand practical scaling properties, while clearer explanation of terminal state handling would aid reproducibility. The trade-offs between computational cost and performance deserve more thorough quantification.

4. Empirical analysis: The method's robustness to different data distributions could be more thoroughly investigated to understand its reliability in varied conditions. The ablation studies could provide deeper insights by more systematically exploring beam width's impact on performance. Additionally, a more detailed analysis of how performance scales with increasing numbers of sub-actions would help understand the method's limitations.

5. Scope of claims: The paper makes several broad claims about BVE's capabilities that could benefit from more nuanced presentation. While the claim about scaling to large action spaces is well supported empirically, the claim about "effectively capturing sub-action dependencies" needs stronger evidence from more diverse types of dependencies. The paper states BVE "outperforms state-of-the-art methods" but this is only demonstrated against two baselines. Similarly, while empirical results suggest reduced overestimation bias, the mechanism and conditions for this reduction could be better explained with theoretical analysis.

**Questions:**

Theory:
- Could you outline what theoretical guarantees might be possible for the tree traversal algorithm's convergence properties?
- How would you characterize the relationship between beam search approximation and policy optimality in your method?
- Could you elaborate on the specific conditions under which BVE would be expected to perform optimally?

Comparative evaluation and claims:
- Your paper addresses similar challenges as "An Investigation of Offline Reinforcement Learning in Factorisable Action Spaces" (https://openreview.net/forum?id=STwxyUfpNV) which relies on a value function decomposition. Have you considered comparing against their adaptations of BCQ, CQL, and IQL?
- Could you evaluate BVE on the DeepMind Control Suite environments with discretised actions (or other complex environments with strongly dependent sub-actions) to validate the broad claim about outperforming state-of-the-art methods?
- What evidence supports the claim about "effectively capturing sub-action dependencies" beyond the current experiments?

Technical implementation and scaling:
- Could you explain how the depth penalty and beam search interact during action selection?
-  What is the computational overhead of tree construction and traversal compared to baselines?
- How are terminal states handled in the tree structure?

Empirical analysis:
- How does the method's performance scale with increasing numbers of sub-actions?
- Could you characterize the method's robustness to different data distributions?
- Could you provide more detailed analysis of how beam width affects the trade-off between performance and computational cost?

---

> ### Author Response · Authors · 2024-11-25
> **Response to Reviewer ahEi Part 1**
>
> Thank you for your thoughtful comments and valuable feedback. We greatly appreciate your recognition of the novelty and practical significance of our approach in tackling an important yet under-explored problem setting. We have carefully addressed your concerns and conducted additional experiments and analyses based on your suggestions.
>
> We believe our responses and the new experiments sufficiently address your concerns, and we kindly request that you consider revising your score. Should any concerns remain after reviewing our responses, we would be grateful for further guidance on how to improve our work to meet your expectations for a higher score. Thank you again for your time and insightful suggestions.
>
> ---
>
> ### W1 Limited comparative evaluation: Particularly relevant is the recent parallel work on factorized action spaces which offers baselines and testing environments
>
> Thank you for bringing this recent work (henceforth [1]) to our attention — we have cited it alongside our existing factored action space baseline [2]. To clarify, [2] introduces a factorized version of BCQ that aligns with the approach described in [1] (as acknowledged in [1] Section 3.2). While we agree that comparing BVE’s performance against other factorized offline RL algorithms is of interest, the results in [1] indicate that factorized BCQ performs comparably to the factorized versions of other offline RL methods.
>
> More critically, the limitations of factorized approaches are well-documented and recognized in both [1] (Sections 4.1 and 4.2) and [2] (Section 3.1). Specifically, factorized action spaces are effective when dependencies among sub-actions are weak and/or reward structures are factorizable. These limitations are algorithm-agnostic, applying to all factorized approaches. In contrast, our objective is to develop an approach that is robust to diverse reward structures (as demonstrated by our experiments in Section 4.1) and explicitly accounts for sub-action dependencies (as demonstrated by our experiments in Section 4.2).
>
> Regarding the environments in [1], the results indicate that sub-actions in these spaces exhibit limited inter-dependence. Specifically, while sub-actions in the DeepMind Control Suite environments require coordination, the optimal value of one sub-action does not strongly depend on the value of another. For example, consider a scenario where applying a torque of 0.5 to the cheetah's back thigh rotor is optimal. The agent must simultaneously learn that, in the state where such a torque is applied, a torque of 0.1 to the cheetah's back shin rotor is also optimal. However, these sub-actions remain independently optimal; their effectiveness does not depend on the presence or absence of the other. While coordination is necessary for achieving optimal performance, the sub-actions themselves are not mutually dependent. This contrasts with several real-world settings. In healthcare, for instance, two medications may be independently effective but, when combined, could produce antagonistic effects. While we acknowledge that BVE is applicable to such spaces, our primary focus is on addressing action spaces where current factorized methods are less effective.
>
>
> ### W2 Theoretical analysis: The paper would benefit from theoretical analysis of convergence properties and formal bounds on beam search approximation error
>
> We would like to clarify that beam search is not used during learning and, therefore, does not influence BVE’s convergence properties. Instead, beam search is applied only during policy extraction, after the value estimates have been learned. Because it is decoupled from the learning process, the beam width can be optimized independently after learning, provided the environment allows for optimization in a manner analogous to fine-tuning. This has been clarified in Section 3.

---

> ### Author Response · Authors · 2024-11-25
> **Response to Reviewer ahEi Part 2**
>
> ### W3.1 Technical details: The paper could better explain the interaction between depth penalty and beam search during action selection
>
> We appreciate the opportunity to clarify the relationship between the depth penalty ($\delta$) and beam search. Both mechanisms are introduced to mitigate errors in action selection caused by inaccurate branch value estimates near the tree root. However, they operate in distinct phases and do not directly interact. Specifically, $\delta$ is applied **during learning** to penalize branch value errors closer to the tree root, prioritizing corrections at higher tree levels where decisions have a broader influence on action selection. Beam search, on the other hand, is used exclusively **after training**, during policy extraction, to facilitate broader exploration of action combinations.
>
> To further investigate the effects of these mechanisms, we conducted two new experiments. First, we extended our original ablation study — which previously assessed only $\delta = 1$ and no depth penalty — to include $\delta$ values of 5, 10, and 15:
>
> | $\mathcal{A}$ | No penalty               | $\delta = 1$              | $\delta = 5$       | $\delta = 10$      | $\delta = 15$      |
> |---------------|--------------------------|---------------------------|--------------------|--------------------|--------------------|
> | 16            | $-95.2 \pm 197.1$        | $\mathbf{-7.5 \pm 2.4}$   | $-124.3 \pm 168.7$ | $-97.2 \pm 91.7$   | $-83.3 \pm 86.2$   |
> | 64            | $-28.8 \pm 37.9$         | $\mathbf{-2.9 \pm 0.2}$   | $-45.0 \pm 37.0$   | $-70.3 \pm 9.5$    | $-70.6 \pm 62.6$   |
> | 256           | $\mathbf{-4.5 \pm 0.4}$  | $-5.6 \pm 1.5$            | $-64.7 \pm 37.2$   | $-38.4 \pm 42.9$   | $-81.8 \pm 54.2$   |
> | 1024          | $\mathbf{-7.6 \pm 3.8}$  | $-7.7 \pm 1.4$.           | $-48.1 \pm 49.5$   | $-474.8 \pm 462.6$ | $-416.0 \pm 426.9$ |
> | 4096          | $-23.5 \pm 28.3$         | $\mathbf{-12.4 \pm 7.9}$  | $-29.2 \pm 37.9$   | $-320.0 \pm 470.0$ | $-811.0 \pm 331.2$ |
> | $\sim$16k     | $-47.4 \pm 61.3$         | $\mathbf{-11.9 \pm 6.9}$  | $-61.9 \pm 56.2$   | $-89.6 \pm 127.2$  | $-545.4 \pm 518.9$ |
> | $\sim$65k     | $\mathbf{-16.6 \pm 4.7}$ | $-19.4 \pm 13.2$          | $-371.5 \pm 512.0$ | $-875.5 \pm 348.1$ | $-1116.8 \pm 30.0$ |
> | $\sim$260k    | $-53.3 \pm 56.0$         | $\mathbf{-12.9 \pm 3.0}$  | $-110.4 \pm 37.7$  | $-89.7 \pm 60.1$   | $-76.3 \pm 62.6$   |
> | $\sim$1M      | $-163.5 \pm 281.8$       | $\mathbf{-26.3 \pm 14.1}$ | $-78.3 \pm 58.9$   | $-82.2 \pm 64.6$   | $-187.6 \pm 132.4$ |
> | $\sim$4M      | $-71.3 \pm 56.2$         | $\mathbf{-35.9 \pm 28.8}$ | $-123.1 \pm 87.6$  | $-165.2 \pm 150.5$ | $-270.3 \pm 221.6$ |
>
> From this experiment, we observe that while a depth penalty generally improves performance, it should be kept small, with $\delta = 1$ yielding the best results across nearly all environments. We have included this experiment in Section 4 of our paper.
>
> Additionally, we measure the effect of varying beam widths on both return and computational overhead. Due to space constraints, we present results below for the environment with over 4 million action spaces, as it requires the largest tree. However, we have conducted the same experiments for all environments with at least 16,000 actions, and these results have been also be included in Section 4 of our paper.
>
> | Width | Return | Total runtime (seconds) | Steps per second |
> |-------|--------|-------------------------|------------------|
> | 1     | -142.6 | 0.02                    | 79.8             |
> | 5     | -20.6  | 0.06                    | 102.5            |
> | 10    | -19.5  | 0.1                     | 54.6             |
> | 25    | -20.9  | 0.2                     | 27.8             |
> | 50    | -21.5  | 0.4                     | 16.5             |
> | 100   | -24.9  | 0.7                     | 9.7              |
> | 200   | -31.3  | 1.5                     | 5.3              |
> | 300   | -26.6  | 1.9                     | 3.7              |
> | 400   | -26.6  | 2.4                     | 2.9              |
> | 500   | -26.6  | 3.0                     | 2.3              |
>
> Our results indicate that beam width does not need to be very large to achieve optimal performance — typically, a beam width of 10 is sufficient. More importantly, the choice of beam width has minimal practical impact on runtime for most real-world applications, as inference time remains well under one second, even with beam widths far exceeding what is necessary or optimal. These results were obtained using a system with a 10-core CPU, 32 GB of unified memory, and no GPU.

---

> ### Author Response · Authors · 2024-11-25
> **Response to Reviewer ahEi Part 3**
>
> ### W3.2 Technical details: A clearer explanation of terminal state handling would aid reproducibility
>
> In our implementation, terminal states yield a reward of 10, after which the episode terminates, and the agent begins the next iteration in the starting state. Notably, action selection in terminal states is inconsequential to the loss function in Equation 4, as the true Q-value for all terminal states is 0. Therefore, if $s'$ is a terminal state, the update target is simply 10. We have clarified this point in Section 3.
>
> ### W3.3 Technical details: The trade-offs between computational cost and performance deserve more thorough quantification
>
> As detailed in W3.1, we have included a new experiment to quantify the computational overhead associated with beam search. Regarding learning efficiency, Figures 8 and 9 in the original paper show that, except in environments with small action spaces (256 or fewer), BVE requires fewer gradient steps than both IQL and FAS to learn a performant policy.
>
> ### W4.1 Empirical analysis: The method's robustness to different data distributions could be more thoroughly investigated
>
> We have conducted two new sets of experiments to evaluate BVE's robustness to different data distributions. First, we created environments with varying levels of dependence among sub-actions by increasing the number of pits in the environment — more pits require greater coordination between sub-actions. To ensure a viable path from the start state to the goal state, pits were restricted to interior (non-boundary) states, which total $3^N$ in the grid world. Specifically, we generated six new datasets with 328, 656, 1640, 3281, 4921, and 6561 pits in the 8D grid world, corresponding to 5%, 10%, 25%, 50%, 75%, and 100% of interior states being pits.
>
> We selected the 8D world as it offers a balance between action-space complexity (65,536 possible actions per state) and the computational feasibility of generating multiple new datasets.
>
> | Number of pits | BVE                       | FAS                | IQL               |
> |----------------|---------------------------|--------------------|-------------------|
> | 0              | $\mathbf{-8.2 \pm 2.2}$   | $-24.4 \pm 10.4$   | $-28.1 \pm 12.1$  |
> | 328            | $\mathbf{-19.0 \pm 2.3}$  | $-78.4 \pm 99.7$.  | $-85.8 \pm 45.0$  |
> | 656            | $\mathbf{-16.8 \pm 3.3}$  | $-140.4 \pm 177.1$ | $-88.1 \pm 50.1$  |
> | 1640           | $\mathbf{-42.9 \pm 43.3}$ | $-178.6 \pm 212.8$ | $-82.8 \pm 52.0$  |
> | 3281           | $\mathbf{-54.8 \pm 47.9}$ | $-902.6 \pm 274.1$ | $-106.4 \pm 42.8$ |
> | 4921           | $\mathbf{-42.9 \pm 34.3}$ | $-942.8 \pm 278.1$ | $-110.0 \pm 34.3$ |
> | 6561           | $\mathbf{-41.6 \pm 22.4}$ | $-1131.4 \pm 0.0$  | $-112.4 \pm 22.8$ |
>
>
> As shown in the table above, BVE's performance remains relatively stable across all environments, whereas FAS's performance drops sharply when half the interior states are pits. Similarly, IQL's performance deteriorates as the number of pits — and consequently, the dependencies among sub-actions — increases.
>
> Second, as described in Section 4 of our paper, we use an augmented form of A$^*$ to generate our datasets. In our original experiments, the A$^*$ agent selects the optimal action with a probability of 0.1, choosing randomly otherwise. To extend this analysis, we created three new datasets for the 8D world with 3,281 pits. These datasets were generated by an A$^*$ agent selecting the optimal action with probabilities of 0.2, 0.3, and 0.4, respectively. We selected the 8D world with 3,281 pits for these experiments because, as shown in the table above, it is the environment that poses the greatest challenge for BVE, making success in this setting more likely to generalize to environments where BVE performs better.
>
> | Suboptimal % | BVE                        | FAS                | IQL               |
> |--------------|----------------------------|--------------------|-------------------|
> | 80           | $\mathbf{-50.6 \pm 44.2}$  | $-576.7 \pm 407.4$ | $-72.5 \pm 51.3$  |
> | 70           | $\mathbf{-54.9 \pm 43.2}$  | $-469.3 \pm 323.2$ | $-65.2 \pm 44.7$  |
> | 60           | $\mathbf{-57.4 \pm 52.1}$  | $-378.8 \pm 258.5$ | $-62.0 \pm 43.7$  |
>
> As can be seen in the above table, BVE consistently outperforms both IQL and FAS across all environments. However, as the proportion of optimal actions in the dataset increases, IQL's performance approaches that of BVE. This is expected given the short time horizon of the problem setting, where the expected number of optimal demonstrations per state is 1,000 when the suboptimal rate is 0.4.
>
> These experiments have been included in Section 4 of our paper.

---

> ### Author Response · Authors · 2024-11-25
> **Response to Reviewer ahEi Part 4**
>
> ### W4.2 Empirical analysis: The ablation studies could provide deeper insights by more systematically exploring beam width's impact on performance
>
> As described in W3.1, we have added this ablation study in response to the reviewer's recommendation.
>
>
> ### W4.3 Empirical analysis: A more detailed analysis of how performance scales with increasing numbers of sub-actions would help understand the method's limitations
>
> We agree that exploring how many sub-actions BVE can scale to would be valuable. However, our original experiments already demonstrate its efficacy in an environment with over 4 million possible actions and more than 48 million states. Notably, in addition to outperforming the baselines across all environments, BVE exhibits only a moderate performance drop as complexity increases.
>
>
> ### W5.1 Scope of claims: The paper states BVE "outperforms state-of-the-art methods" but this is only demonstrated against two baselines
>
> We acknowledge that including additional baselines would further strengthen the demonstration of BVE's efficacy. However, to the best of our knowledge, factorization is the only approach specifically designed for offline learning in discrete, combinatorial action spaces. While it is possible to include additional factorized baselines, as discussed in W1, the limitations of value function decomposition are well-documented, and empirical evidence in [1] indicates that factorized BCQ (FAS in our experiments) performs comparably to other factorized offline methods.
>
>
> ### W5.2 Scope of claims: While empirical results suggest reduced overestimation bias, the mechanism and conditions for this reduction could be better explained with theoretical analysis
>
> We appreciate the opportunity to clarify how BVE leverages well-established offline RL principles to address overestimation bias.
>
> The first mechanism is the behavior cloning term $\|\hat{\mathbf{a}}' - \mathbf{a}'\|$ in Equation 4, which encourages the agent to select actions similar to those chosen by the behavior policy. This term can be viewed as a value-based analog to the behavior cloning regularizer introduced in TD3+BC (Equation 3). Despite its simplicity, such regularization has been shown to be effective across diverse problem settings [3].
>
> The second mechanism is the sparsification of the action tree, restricting Q-value evaluations to actions present in the dataset. This approach mitigates overestimation bias by avoiding erroneous Q-values caused by extrapolation error, as described in [4] (BCQ) and [5]. In BCQ, this restriction is achieved through a generative model, whereas BVE accomplishes the same effect by sparsifying the action tree.
>
>
> ### Q1.1 Theory: Could you outline what theoretical guarantees might be possible for the tree traversal algorithm's convergence properties?
>
> In the tabular case, BVE converges with the same guarantees as tabular Q-learning [6]. While we agree that a rigorous theoretical analysis of BVE's convergence properties under function approximation would be valuable, we consider it beyond the scope of the current work. This is an important direction for future research, as it could build upon the foundation established here.
>
>
> ### Q1.2 Theory: How would you characterize the relationship between beam search approximation and policy optimality in your method?
>
> As described in W3.1, beam search does not influence the learned Q-values; it is used exclusively during policy extraction. As demonstrated in W3.1, we conducted additional experiments to evaluate the impact of beam search on the quality of the extracted policy. Furthermore, as noted in W2, since beam search is decoupled from the learning process, the beam width can be optimized independently after learning, provided the environment permits optimization in a manner analogous to fine-tuning.
>
>
> ### Q1.3 Theory: Could you elaborate on the specific conditions under which BVE would be expected to perform optimally?
>
> A key condition for BVE's effectiveness is an environment with a combinatorial, discrete action space, as such spaces align naturally with the tree structure central to BVE's operation. However, we believe BVE's true strength lies in environments where sub-actions are strongly dependent or reward functions are not factorizable. By explicitly accounting for sub-action dependencies, BVE can learn highly performant policies in these challenging settings, where current state-of-the-art methods are less effective [1, 2].
>
>
> ### Q2.1 Comparative evaluation and claims: Have you considered comparing against factorized adaptations of BCQ, CQL, and IQL?
>
> As detailed in W1, our FAS baseline is a factorized adaptation of BCQ. Given the reported similarity in performance among factorized variants of offline RL methods in [1], as well as the known limitations of factorized methods described in [1] (Sections 4.1 and 4.2) and [2] (Section 3.1), we believe our experiments sufficiently demonstrate the strengths of BVE relative to factorized methods.

---

> ### Author Response · Authors · 2024-11-25
> **Response to Reviewer ahEi Part 5**
>
> ### Q2.2 Comparative evaluation and claims: Could you evaluate BVE on the DeepMind Control Suite environments with discretised actions
>
> As discussed in W1, while we agree that BVE is applicable to such settings, our primary focus is on environments with non-factorizable reward structures and/or strong dependencies among sub-actions, which are common in real-world scenarios. The results reported in [1] suggest that sub-actions in the DeepMind Control Suite environments exhibit limited interdependence. Specifically, while sub-actions in these environments require coordination, the value of one sub-action does not strongly depend on the value of another.
>
> For instance, consider applying a torque of 0.5 to the cheetah's back thigh rotor. In parallel, the agent must learn that, in the state where such a torque is applied, a torque of 0.1 is optimal for the back shin rotor. However, these sub-actions are independently optimal; their effectiveness is not conditional on the presence or absence of the other. This contrasts with several real-world settings. For example, in healthcare, two medications might be independently effective but could produce antagonistic effects when combined.
>
>
> ### Q2.3 Comparative evaluation and claims: What evidence supports the claim about "effectively capturing sub-action dependencies" beyond the current experiments?
>
> As described in W4.1, we conducted additional experiments to evaluate BVE's robustness to varying strengths of sub-action dependencies. Notably, in the 8D environment with 6,561 pits, all interior states are pits. In this setting, failure to perfectly account for sub-action dependencies leads to the agent falling into a pit, resulting in catastrophic failure.
>
>
> ### Q3.1 Technical implementation and scaling: Could you explain how the depth penalty and beam search interact during action selection?
>
> As described in W3.1, these mechanisms are designed to reduce errors in action selection caused by inaccurate branch value estimations near the tree root. However, they do not directly interact, as they are applied in different phases. Specifically, the depth penalty is used **during learning** to penalize branch value errors closer to the tree root, prioritizing corrections at higher levels of the tree where decisions have a broader impact on the selected action. Beam search, on the other hand, is used exclusively **after training**, during policy extraction, to enable broader exploration of action combinations. We have clarified this point in Section 3.
>
>
> ### Q3.2 Technical implementation and scaling: What is the computational overhead of tree construction and traversal compared to baselines?
>
> In our experiments, tree construction occurs dynamically. Specifically, when performing branch value estimation, as detailed in Algorithm 1, a node's children are computed at runtime. Depending on the size of the tree and the nature of the environment, these values could be cached or the entire tree precomputed. However, we note that generating children at runtime is computationally inexpensive.
>
> As discussed in W3.1, we conducted an additional experiment to measure the impact of different beam widths on returns and computation time. During learning, tree traversal corresponds to a beam width of 1, which, as shown in the results in W3.1, is a computationally efficient procedure.
>
>
> ### Q3.3 Technical implementation and scaling: How are terminal states handled in the tree structure?
>
> As detailed in W3.2, if $s'$ is a terminal state, the update target is equal to the reward, since the true Q-value of a terminal state is always 0.
>
>
> ### Q4.1 Empirical analysis: How does the method's performance scale with increasing numbers of sub-actions?
>
> As detailed in W4.3, BVE scales to environments with over 4 million sub-actions and 48 million states. Notably, in addition to outperforming the baselines across all environments, BVE experiences only a moderate performance drop from the simplest to the most complex environment.
>
>
> ### Q4.2 Empirical analysis: Could you characterize the method's robustness to different data distributions?
>
> As detailed in our response to W4.1, we present results from two new sets of experiments demonstrating BVE's robustness to data distributions with varying strengths of sub-action dependencies and levels of dataset expertise.
>
> ### Q4.3 Empirical analysis: Could you provide more detailed analysis of how beam width affects the trade-off between performance and computational cost?
>
> As detailed in W3.1, we have empirically investigated this trade-off through a new set of experiments. Our findings show that beam widths do not need to be very large to achieve optimal performance — typically, a width of 10 is sufficient. Furthermore, the choice of beam width has minimal impact on runtime in most real-world applications, as inference time remains well under a second, even with beam widths far exceeding what is necessary or optimal, including when using a processor without a GPU.

---

> ### Author Response · Authors · 2024-11-25
> **References Cited in Response to Reviewer ahEi**
>
> [1] Alex Beeson, David Ireland, and Giovanni Montana. 2024. An Investigation of Offline Reinforcement Learning in Factorisable Action Spaces. Transactions on Machine Learning Research.
>
> [2] Shengpu Tang, Maggie Makar, Michael W. Sjoding, Finale Doshi-Velez, and Jenna Wiens. 2023. Leveraging Factored Action Spaces for Efficient Offline Reinforcement Learning in Healthcare (2023).
>
> [3] Scott Fujimoto and Shixiang Shane Gu. 2021. A minimalist approach to offline reinforcement learning. Advances in neural information processing systems 34, (2021), 20132–20145.
>
> [4] Scott Fujimoto, David Meger, and Doina Precup. 2019. Off-policy deep reinforcement learning without exploration. In International conference on machine learning, PMLR, 2052–2062.
>
> [5] Sergey Levine, Aviral Kumar, George Tucker, and Justin Fu. 2020. Offline reinforcement learning: Tutorial, review, and perspectives on open problems. arXiv preprint arXiv:2005.01643 (2020).
>
> [6] Richard S. Sutton and Andrew G. Barto. 2018. Reinforcement Learning: An Introduction. A Bradford Book, Cambridge, MA, USA.

---

> ### Author Response · Authors · 2024-12-01
> **Follow up on rebuttal**
>
> We sincerely appreciate your feedback and your support of our work! We would like to remind you that we have addressed your concerns. As the discussion period nears its conclusion, please let us know if there are any remaining points we can address within the timeframe.

---

### Author Response · Authors · 2024-11-25
**Author Rebuttal**

We sincerely thank the reviewers for their insightful feedback and their recognition of our contributions. Reviewers emphasized the importance of the studied problem setting, noting that **"the paper targets an important problem"** [oUY2] that **"has indeed been underexplored, making it worthy of further attention"** [8i81]. They highlighted the novelty of our approach, stating that **"the paper introduces a novel way to represent combinatorial action spaces using trees"**, **"the combination of beam search with RL is innovative and hadn't been applied this way before"**, **"the method provides a new angle on addressing overestimation bias in offline RL"** [ahEi], and **"the application of tree-structured action search seems new"** [oUY2]. Reviewers also commended the experimental rigor, remarking that **"the experimental design directly addresses the key claims"** [ahEi], **"experiments presented are also quite persuasive and effectively demonstrate the advantages of BVE"** [8i81], and that **"the ablation studies effectively isolate the contribution of each component"** [ahEi]. Furthermore, they praised the paper's clarity and readability, observing that **"the paper explains its main idea clear"** [oUY2], **"the algorithm descriptions are precise and well-detailed with clear pseudocode"** [ahEi], and **"complex concepts are explained through effective visualizations and examples"** [ahEi].

Reviewers raised several important points, which we address comprehensively in our point-by-point responses. In this general remark, we summarize the new experiments conducted during the rebuttal period and highlight key topics emphasized by the reviewers.

Based on the reviewers’ feedback, we conducted five new sets of experiments to:

- Analyze the effect of sub-action dependency strength on BVE's performance
- Evaluate BVE's robustness across varying data distributions
- Test BVE's efficacy in online fine-tuning following offline RL
- Assess the impact of beam width on policy performance and runtime
- Examine the effect of varying depth penalties

These additional experiments and analyses, together the initial results, further strengthen our confidence in BVE's effectiveness.

We address the key points raised by the reviewers as follows:

- We demonstrate that BVE effectively captures sub-action dependencies, even in settings where each action requires accounting for dependencies among all sub-actions to avoid catastrophic failure [ahEi, 8i81].
- We clarify how BVE mitigates overestimation bias through its tree structure and loss function [ahEi, oUY2, 8i81].
- We justify the use of BVE for offline settings and emphasize that its ability to be used for online fine-tuning after offline RL is a significant strength, similar to that of IQL, which is also designed for offline RL [oUY2, 8i81].

### General Notes to Reviewers:

Please note that we use W# to address weaknesses and Q# to address questions in our responses.

We sincerely thank the reviewers for their thoughtful comments, which have significantly strengthened our paper. If there are any remaining concerns that you feel we have not adequately addressed to warrant an increased score, we would greatly appreciate further clarification and suggestions for improvement. Thank you once again for your time and valuable reviews.

---

### Meta-Review · Area_Chair_kPTX · 2024-12-23

**Metareview:**

The paper describes a new RL technique for problems with combinatorial action spaces.  It claims that it can scale to problems with high number of sub actions (which induce a joint action space that is exponential in the number of sub actions).  The strengths of the paper include the importance of the problem tackled, an interesting idea and comprehensive experiments.  The weaknesses include a lack of clarity regarding the algorithm, how it handles interdependencies between sub-actions, how it prevents overestimation in offline training and how the scalability achieved impacts accuracy.  Since this lack of clarity is core to the proposed algorithm, the paper is not ready for publication.

**Additional Comments On Reviewer Discussion:**

Unfortunately, the reviewers did not provide any feedback during the rebuttal period and did not answer the AC's request for discussion.  The AC read the paper, the reviews and the responses.  The AC commends the authors for improving the paper substantially.  Most of the concerns of the reviewers were addressed in the revised manuscript and the rebuttal clarified many aspects.  However, the description of the proposed algorithm remains unclear as well as how the approach handles interdependencies between sub-actions, how overestimation is prevented in offline training and how scalability impacts accuracy.  Let me explain.

The paper explains that Q-values will be organized in the form of a tree to achieve scalability.  However, it is not clear how to organize the tree.  For instance, the tree in Fig 1 shows that the nodes in the first layer correspond to different instantiations of the first sub action.  How was it determined that we should consider different instantiations of the first sub action instead of the other sub actions.  The second layer corresponds to different instantiations of the second and third sub action.  Why consider instantiations of two sub actions simultaneously instead of one sub action?  Node [1,1,1] is included as a child of node [1,1,0].  Why not include [1,1,1] as a child of node [1,0,1]?  The paper does not explain how those choices were made and generally how to construct the tree.

Ultimately, the tree must include all exponentially many joint actions as nodes.  So technically there is no reduction in complexity due to the tree itself.  The reduction in complexity arises from the greedy traversal of the tree where only one branch is expanded at decision time.  This is fine as long as the tree is accurate.  However, since the tree includes all joint actions, the tree will at best provide approximate Q-values.  The accuracy of the tree will depend on the training.  Since it is impossible to visit all combinations of state-joint-action pairs, then the tree cannot be accurate.  In fact the accuracy of the tree will depend a lot on the architecture of the neural net used to predict the Q-values.  The paper claims that by considering joint actions, it can automatically handle interdependencies between sub actions.  This is insufficient.  The neural network needs to have a suitable structure that induces the right inductive bias to capture interdependencies between sub actions since it is not possible to train by visiting all state-joint-action pairs.  The paper needs to justify its choice of neural architecture and why it should capture sub action interdependencies well.

The paper claims that it prevents overestimation in offline training by utilizing a behavioural cloning regularizer similar to previous work.  This is indeed the case for the leaf nodes in the tree as shown in Line 3 of Algorithm 1.  However, there is no regularization for the loss computation of the other nodes of the tree as shown in Line 12 of Algorithm 1.  In fact, Line 15 of Algorithm 1 computes a max that will lead to overestimation due to the absence of any regularization.  This needs to be discussed.

In the rebuttal, the authors claim that the tree does not impact the convergence of the algorithm because the tree is used only at decision time.  However, the tree is also used during training to compute the BVE loss.  Hence it is not clear how the tree impacts convergence.  This needs to be discussed in the paper.

Overall, the approach is promising, but its description and analysis require more work before publication.

---

### Decision · Program_Chairs · 2025-01-22

Reject